# Real-time visualisation of the intracellular dynamics of conjugative plasmid transfer

Agathe Couturier[1], Chloé Virolle[1], Kelly Goldlust[1], Annick Berne-Dedieu[1], Audrey Reuter[1], Sophie Nolivos [1], Yoshiharu Yamaichi [2], Sarah Bigot [1] ✉ & Christian Lesterlin [1] ✉

Conjugation is a contact-dependent mechanism for the transfer of plasmid DNA between bacterial cells, which contributes to the dissemination of antibiotic resistance. Here, we use live-cell microscopy to visualise the intracellular dynamics of conjugative transfer of F-plasmid in *E. coli*, in real time. We show that the transfer of plasmid in single-stranded form (ssDNA) and its subsequent conversion into double-stranded DNA (dsDNA) are fast and efficient processes that occur with specific timing and subcellular localisation. Notably, the ssDNA-to-dsDNA conversion determines the timing of plasmid-encoded protein production. The leading region that first enters the recipient cell carries single-stranded promoters that allow the early and transient synthesis of leading proteins immediately upon entry of the ssDNA plasmid. The subsequent conversion into dsDNA turns off leading gene expression, and activates the expression of other plasmid genes under the control of conventional double-stranded promoters. This molecular strategy allows for the timely production of factors sequentially involved in establishing, maintaining and disseminating the plasmid.

Bacterial DNA conjugation is a widespread horizontal gene transfer mechanism in which genetic information is transmitted from a donor to a recipient cell by direct contact[1–4]. Conjugation is responsible for the intra- and inter-species dissemination of various metabolic properties and accounts for 80% of acquired resistances in bacteria[5]. The F plasmid was the first conjugative element discovered[1,6] and is now documented as the paradigmatic representative of a large group of conjugative plasmids widespread in *Escherichia coli* and other Enterobacteriaceae species, in which they are associated with the dissemination of colicins, virulence factors and antibiotic resistance[7–9]. Due to their fundamental and clinical importance, F-like plasmids have been the focus of extensive studies that provided a detailed understanding of the molecular reactions and factors involved in their transfer by conjugation[3,4].

Within the donor cell, the relaxosome components, including the integration host factor IHF, plasmid-encoded accessory proteins TraY, TraM and the multifunctional relaxase TraI, are recruited to the origin

of transfer (*oriT*) of the F plasmid[10–12]. The relaxosome complex is then recruited to the Type IV secretion system (T4SS) by the coupling protein TraD, resulting in the formation of the pre-initiation complex[13–17]. TraI and TraD proteins are archetype components of the core set of subunits required for the establishment of active conjugation machinery and are respectively referred to as VirD2 and VirD4 in the common nomenclature. It is proposed that the establishment of the mating pair induces a still uncharacterised signal that activates the pre-initiation complex. Then, TraI introduces a site- and strand-specific DNA cut (nick) into the plasmid's *oriT* and remains covalently bound to the 5′ phosphate end. TraI also serves as a helicase that extrudes the ssDNA plasmid to be transferred, called the T-strand[18–26]. It was initially suggested and later confirmed that two relaxases are required to carry out these functions[27,28]. At this stage, the 3′OH of the T-strand serves to initiate the rolling-circle replication (RCR) that converts the intact circular ssDNA plasmid into dsDNA in the donor cell[3,29,30], while the 5′phosphate bound to TraI is transferred into the recipient cell through

[1]Molecular Microbiology and Structural Biochemistry (MMSB), Université Lyon 1, CNRS, Inserm, UMR5086, 69007 Lyon, France. [2]Université Paris-Saclay, CEA, CNRS, Institute for Integrative Biology of the Cell (I2BC), 91198 Gif-sur-Yvette, France. ✉e-mail: Sarah.Bigot@ibcp.fr; Christian.lesterlin@ibcp.fr

the T4SS machinery. While the molecular structure of the T4SS has been well characterised[31–34], the way the T-complex (T-strand-TraI nucleoprotein) is translocated through the membrane of the donor and recipient cells' membranes remains unclear.

The first transferred segment is the ~13.5 knt leading region, carrying genes which encode the Ssb[F] protein homologue to the chromosomally encoded essential single-strand-binding protein Ssb, the PsiB protein (Plasmid SOS Inhibition)[35–38] that inhibits SOS induction during conjugation[39,40], and other proteins of unknown function. Remarkably, the leading region is conserved in various enterobacterial plasmids belonging to a variety of incompatibility groups[41–46]. The adjacent and next transferred ~17 knt maintenance region carries the ParABS-like plasmid partition system (SopABC) and the origins of vegetative replication[47–50]. The last transferred segment of the F plasmid is the large ~33.3 knt *tra* region that encodes all the protein factors required for plasmid DNA processing and transfer, including the relaxosome, the T4SS and the exclusion system against self-transfer[4]. Besides, F-like plasmids often carry cargo genes involved in various metabolic functions commonly integrated between the maintenance and the *tra* regions[7,9]. Once both the 5' and the 3' ends of the T-strand have been internalised into the recipient cell, now called a transconjugant, the ssDNA plasmid is circularised by TraI[26,27,51,52]. The ssDNA plasmid will also be converted into dsDNA by the complementary strand synthesis reaction. Whether this DNA synthesis reaction occurs as the plasmid enters the recipient cell or is initiated after plasmid recircularization remains unclear. Nonetheless, the completion of the ss-to-dsDNA conversion is required for plasmid replication and partition and is, therefore, critical to plasmid stability in the new host cell lineage.

The above-described mechanistic model is well-documented; however, the real-time dynamics and intracellular organisation of conjugation remain largely undescribed in the live bacterium. In particular, we know very little about the subcellular localisation and timing of the reactions in the recipient cell, including the ssDNA plasmid entry, the ss-to-dsDNA conversion and plasmid gene expression. Regarding the last-mentioned, early works reported that some leading genes (*ssb[F]* and *psiB* in F plasmid, and *ssb[Collb-P9]*, *psiB* and *ardA* in Collb-P9 plasmid) are expressed rapidly after entry of the plasmid in the acceptor cell[36–38,42,53,54]. In vitro work by Masai *et al.*[55] showed that the single-stranded form of the non-coding F*rpo* sequence, located in the F plasmid leading region, folds into a stem-loop structure that reconstitutes canonical −10 and −35 boxes. This promoter sequence can recruit the *E. coli* RNA polymerase that initiates RNA synthesis in in vitro assays[55]. Sequences homologous to F*rpo* were also found in the leading region of Collb-P9[56,57]. These observations led to the proposal that F*rpo*-like sequences could act as ssDNA promoters initiating the early transcription of leading genes when the plasmid is still in ssDNA form. Whether this regulation mechanism happens during in vivo conjugation remains to be demonstrated.

In this study, we use live-cell microscopy imaging to visualise the complete transfer sequence of the native F plasmid between *E. coli* K12 strains. We inspect the key steps of conjugation using specifically developed genetic reporters, including a fluorescent fusion of the chromosomally encoded single-strand-binding protein Ssb (Ssb-Ypet) to monitor the ssDNA transfer, the mCherry-ParB/*parS* system to reveal the ss-to-dsDNA conversion and subsequent plasmid duplication, and translational fluorescent fusions to quantify and time plasmid-encoded production in the new host cell[58,59]. This approach uncovers the choreography of conjugation reactions in live bacteria and provides new insights into the interplay between plasmid processing and gene expression.

## Results

### Dynamics of the ssDNA plasmid during transfer

We monitored the dynamic localisation of a fluorescent endogenous fusion of the chromosomally encoded single-strand-binding protein Ssb (Ssb-Ypet) in donor and recipient cells, during vegetative growth and conjugation (Fig. 1a, b and Fig. S1). During vegetative growth, Ssb-Ypet forms discrete foci at midcell and quarter positions within the inner region of donors and recipient cells (Fig. 1c and Fig. S2a, b). These Ssb foci, termed Ssb replicative foci hereafter, are associated with the ssDNA that follows the replication forks onto the nucleoid DNA[60,61]. During conjugation, the intracellular localisation of Ssb changes dramatically. As previously reported[58,59], the entry of the ssDNA plasmid in the recipient cell, now called a transconjugant, triggers the recruitment of Ssb molecules and the formation of bright membrane-proximal foci, we termed Ssb conjugative foci (Fig. 1b and Fig. S1). Here, we also observe the formation of Ssb conjugative foci in the donor cells, thus revealing the presence of ssDNA plasmid on each side of the conjugation pore during transfer (Fig. 1b and Fig. S1). Foci localisation analysis reveals that plasmid exit and entry occur at specific membrane positions within the mating pair cells. Ssb conjugative foci are distributed at the periphery of the donor cell and preferentially at the quarter positions (Fig. 1c and Fig. S2a, b), reflecting the preferred position for the exit of the ssDNA plasmid through active conjugation pores. By contrast, ssDNA plasmid entry predominantly occurs within the polar regions of the transconjugant cells (Fig. 1c and Fig. S2a, b). Our data also allow us to address whether conjugation occurs at a specific cell cycle stage. Analysis of cell length as a proxy of cell age reveals that donor and recipient cells engaged in plasmid transfer exhibit similar length distribution than during vegetative growth (Fig. 1d). This shows that the donors can give, and recipients can acquire the plasmid at any stage of their cell cycle, from birth to cell division.

In 77.8 ± 7% (n = 131) of individual plasmid transfer events visualised by time-lapse imaging (1 min/frame), Ssb conjugative foci appear in the donor and transconjugant cells on the same frame (Fig. 1e). In these cases, Ssb conjugative foci are, on average brighter in the transconjugant than in the donor cells, reflecting the relative amount of ssDNA plasmid on each side of the conjugation pore (Fig. 1f). In the remaining 22.2% of transfer events, Ssb conjugative foci first appear in the transconjugant and then in the donor one or two minutes later (Fig. 1e). The delayed accumulation of ssDNA in the donor relative to the recipient is corroborated by the quantification of a 2.9 ± 1.1 min (n = 294) average lifespan of Ssb-Ypet conjugative foci in the transconjugants, compared to 2.5 ± 1.1 min (n = 197) in the donor cells (Fig. 1g). These data indicate that the appearance of conjugative foci is asynchronous in the mating pair cells and suggest a specific sequence of ssDNA transfer. The first segment of the T-strand generated by the helicase activity of TraI in the donor cell does not dwell long enough to recruit Ssb molecules and is immediately transferred to the recipient. Only after this brief transfer stage does the ssDNA accumulates on the donor's side as well, where it can correspond to either or both the non-transferred plasmid strand or to the T-strand. This implies that the rate of ssDNA formation by TraI helicase activity is faster than that of ssDNA removal by the RCR and transfer through the T4SS (See discussion).

The internalisation of a large amount of ssDNA plasmid provokes the massive recruitment of the intracellular pool of Ssb molecules at the periphery of the donor and transconjugant cells. This change in Ssb-Ypet subcellular distribution is revealed by skewness analysis, which provides a non-biased measure of the asymmetry of fluorescence distribution within the cells without a requirement for threshold-based foci detection (Fig. 1h). Wild-type cells producing a free mCherry (mCh) exhibit a low skewness corresponding to the homogeneous pixel fluorescence distribution inside the cell's cytoplasm. During vegetative growth, Ssb-Ypet fluorescence is partly diffuse in the cytoplasm and partly locally concentrated within replicative foci, resulting in skewness of ~1.2. By comparison, Ssb-Ypet exhibits a strong skewness of ~4.1 in donors and transconjugants during plasmid transfer, reflecting the increased proportion of Ssb molecules clustered within foci. Hence,

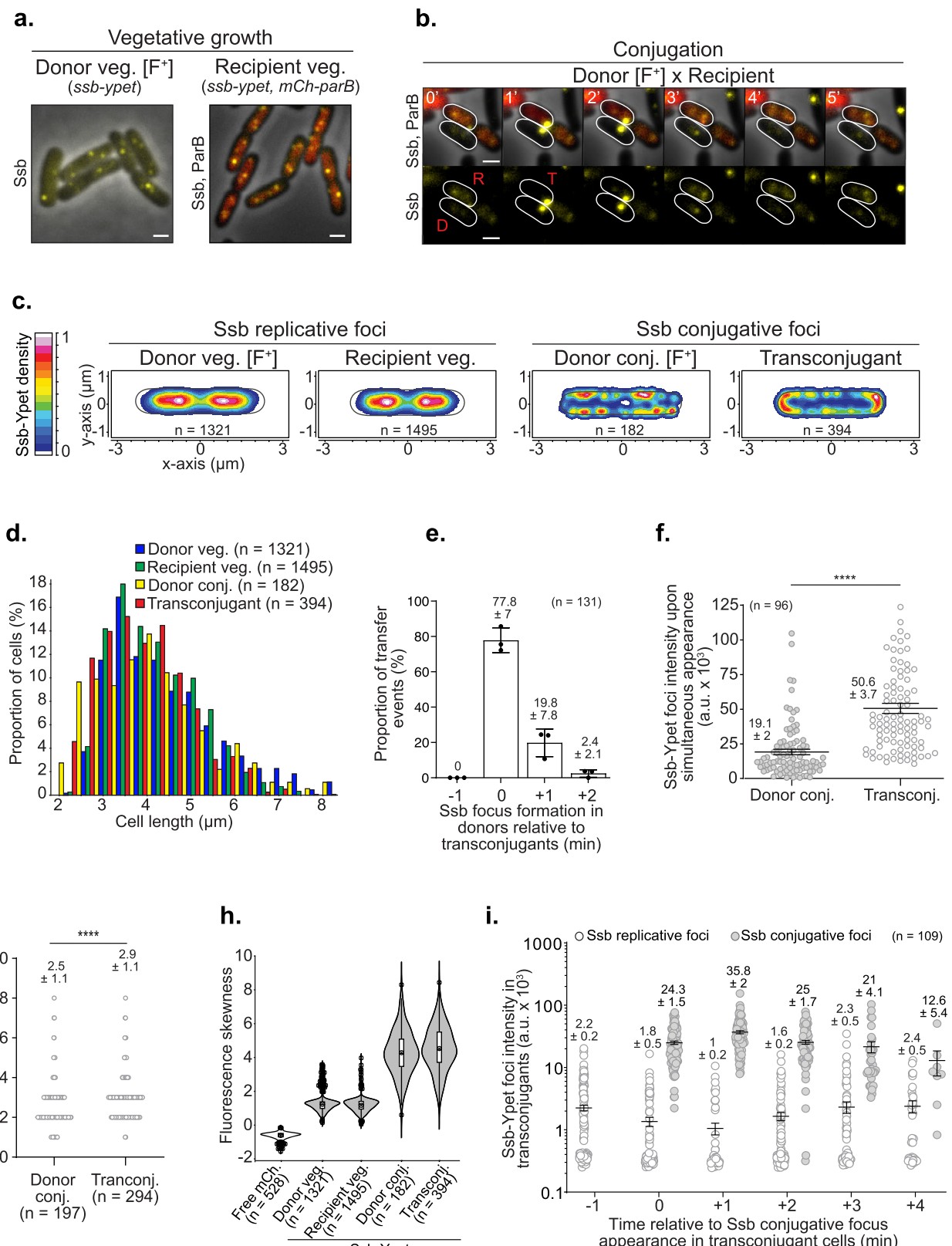

we wondered what part of Ssb molecules are contained within conjugative foci and if their formation was associated with a depletion of Ssb within replicative foci in the transconjugant cell. To address this question, we performed Ssb-Ypet foci automatic detection and brightness quantification during plasmid transfer (Fig. 1i). We observe that one minute after the beginning of plasmid

entry Ssb-Ypet replicative foci are still present but exhibit half their initial intensity, while conjugative foci are 35 times brighter. Since the total Ssb-Ypet intracellular fluorescence is unchanged during the transfer (Fig. S2c), these variations can be attributable to the displacement of Ssb-Ypet molecules onto the incoming ssDNA plasmid rather than Ssb-Ypet de novo synthesis. This dynamic

**Fig. 1 | Real-time dynamics of ssDNA plasmid transfer from donor to recipient cells. a** Representative microscopy images of donors and recipients carrying the *ssb-ypet* fusion gene during vegetative growth. The recipients also produce the diffuse mCh-ParB fluorescent protein. Scale bars 1 μm. **b** Time-lapse microscopy images of plasmid transfer between a donor (D) and a recipient (R) that is converted into a transconjugant cell (T). Scale bars 1 μm. Additional events are presented in Figure S1. **c** 2D localisation heatmaps of Ssb-Ypet in donors, recipients and transconjugant during vegetative growth (veg.) and conjugation (conj.). Normalisation by the cell length of (*n*) individual cells from at least three biological replicates. The density scale bar is on the left. **d** Cell length distribution histogram of donors, recipients and transconjugants (*n* cells analysed from at least three independent experiments). **e** Ssb conjugative focus appearance timing in donor relative to transconjugant cells. Histograms with means and SD represent the proportion of transfer events in which the Ssb focus appears in the donors before (−1 min), simultaneously (0 min) or after (+1 min; +2 min) it appears in transconjugants. The number (*n*) of individual transfer events analysed from three independent experiments is indicated. **f** Jitter plot of the fluorescence intensity of Ssb-Ypet conjugative foci upon simultaneous appearance. The number of foci analysed from three independent experiments (*n*) is indicated with the corresponding Mean and SEM. *P* value significance from Mann–Whitney two-sided statistical test is indicated by ****(*P* ≤ 0.0001). **g** Jitter plots of Ssb-Ypet conjugative foci lifespan in donor and transconjugant cells. *P* value significance from Mann–Whitney two-sided statistical test is indicated by ****(*P* = 0.0001). The number (*n*) of cells analysed from at least five independent experiments is indicated. **h** Violin plots of the fluorescence skewness of a free mCherry and of the Ssb-Ypet in donors, recipients and transconjugant cells. The median, quartile 1 and quartile 3 are indicated by the boxes' bounds, the mean by a black dot, and the minima and maxima by the whiskers' limits. Black dots above and below the max and min values correspond to outliers. Free mCherry data correspond to one representative experiment. Other plots correspond to the same data set as in panel (**c**) from at least three biological replicates. The number of cells analysed (*n*) is indicated. **i** Jitter plot of Ssb-Ypet replicative and conjugative foci intensity in transconjugant cells during conjugation. Time 0 min corresponds to the appearance of the Ssb-Ypet conjugative focus in recipients. The number of cells analysed (*n*) from three independent experiments is indicated with the corresponding Mean and SEM. Donor (LY1007), recipient (LY358), transconjugant (LY358 after F*wt* acquisition from LY1007); the free mCherry is produced from the chromosome in MS388 *wt* background (LY1737). Source data are provided as a Source Data file.

reflects that the incoming ssDNA plasmid recruits most Ssb-Ypet molecules in the acceptor cell during transfer.

It has been estimated that Ssb is present at about ~1320 ± 420 monomers per *E. coli* cell and that a dimer of tetramers covers about 170 nt in vivo[61]. Consequently, there are not enough Ssb copies per cell to accommodate the 108,000 nucleotides of ssDNA F plasmid, plus the few hundreds of nucleotides of ssDNA associated with replication forks (~650 nt at 22 °C[62]). In fact, it is not known whether the F plasmid is ever fully present in ssDNA form in the recipient, as it is not known if the complementary strand synthesis reaction occurs concomitantly with or after the completion of the T-strand transfer. Still, the observed massive recruitment of Ssb molecules onto the incoming ssDNA could reduce Ssb availability and provoke a transitory disturbance of the host chromosome DNA replication. One way to address this question in vivo is to monitor a fluorescent fusion of the $\beta_2$-clamp replisome component (mCh-DnaN), which is diffused in the cytoplasm of non-replicating cells and forms discrete replisome-associated foci during DNA replication progression[60,61,63]. Microscopy imaging and skewness analysis showed no change in DnaN localisation pattern before, during or after Ssb conjugative foci formation (Fig. S2d). This indicates that Ssb recruitment onto the incoming ssDNA plasmid does not result in the collapse of the replication fork. Whether the rate of DNA replication is affected during this transient and short process remains a possibility.

## ss-to-dsDNA conversion and subsequent plasmid replication in the transconjugant cells

The conversion of the newly acquired ssDNA plasmid into dsDNA by the complementary strand synthesis reaction and the subsequent plasmid duplication events were analysed using the *parS*/ParB DNA labelling system[58,59]. The *parS* binding site is inserted in the F plasmid, while the ParB binding protein fluorescently labelled with the mCherry (mCh-ParB) is produced from a plasmid in recipient cells only. Under the microscope, the ss-to-dsDNA conversion is reported by the disappearance of the Ssb-Ypet conjugative focus and the formation of an mCh-ParB focus in the transconjugant cells (Fig. 2a). We first performed time-lapse imaging (1 min/frame) to visualise the success rate and timing of ss-to-dsDNA conversion after ssDNA entry (Fig. 2b). Analysis shows that the appearance of the Ssb-Ypet conjugative focus is followed by the formation of the mCh-ParB focus in 83.3 ± 2.3% (*n* = 311) individual transconjugant cells analysed, indicating that the vast majority of internalised ssDNA plasmids are successfully converted into dsDNA plasmids (Fig. 2c). Notably, we observe that 40 ± 3.2% (*n* = 286) of transconjugant cells where the newly acquired ssDNA plasmid has already been converted into dsDNA subsequently

receive additional ssDNA (Fig. 2d and Fig. S3a). We quantify that 92 ± 3.1% of these multiple ssDNA acquisition events originate from the same donor, among which 79 ± 5.3% appear to take place at the same membrane position, suggesting that they occur through the same conjugation pore (Fig. S3a). The evidence for multiple transfers within an established mating pair demonstrates that a single donor can successively give several copies of the T-strand and that transconjugants in which the ss-to-dsDNA conversion has already been achieved do not become instantly refractory to de novo plasmid acquisition. Accordingly, establishing immunity to conjugation by transconjugant cells is expected to require the production of the plasmid-encoded exclusion proteins TraS and TraT.

Considering successful ss-to-dsDNA events only, we calculate an average 4 ± 1.6 min (*n* = 475) time lag between the appearance of the Ssb-Ypet conjugative focus and the formation of the mCh-ParB focus (Fig. 2e). This period reflects the time required for the completion of a reaction cascade that comprises the internalisation of the ssDNA plasmid, the initiation of the complementary strand synthesis replication, and the recruitment of ParB molecules on the *parS* site in dsDNA form. Though our system does not allow evaluating each step's contribution, results show that the complete sequence of reactions is achieved within a relatively short and consistent period.

Next, we first performed time-lapse imaging (5 min/frame) to examine the timing of plasmid duplication in transconjugant cells (i.e., replication and visual separation of the plasmid copies) (Fig. 2b). We estimate an average of 10.4 ± 4.7 min (*n* = 158) period between the ssDNA-to-dsDNA conversion and the first plasmid duplication event (from one to two mCh-ParB foci) and similar 10.1 ± 5.1 min (*n* = 124) between the first and the second duplication event (from two to three or four mCh-ParB foci) (Fig. 2f). We then decided to compare the rate of plasmid duplication in transconjugants to the rate of plasmid duplication in a vegetatively growing F-carrying donor strain. To do so, we plotted the number of plasmid foci per cell from the ss-to-dsDNA conversion (mCh focus appearance) to cell division in transconjugants and from cell birth to cell division in F-carrying donor cells (Fig. 2g). Results show that the number of F per cell increases significantly faster in transconjugant cells than in vegetatively growing F-carrying cells (75% increase of the fit curve slope), yet to reach a similar final number of ~4 ± 1 copies per cell before division (Fig. 2g). F copy number, like chromosome replication, is known to be controlled by the cell cycle progression, where initiation occurs when a constant mass per origin is achieved[64]. Therefore, our observations are consistent with the interpretation that when a single plasmid copy arrives in a recipient cell that can be at any cell cycle stage, plasmid replication initiation is unrepressed until the specific number of plasmid copies per cell mass is

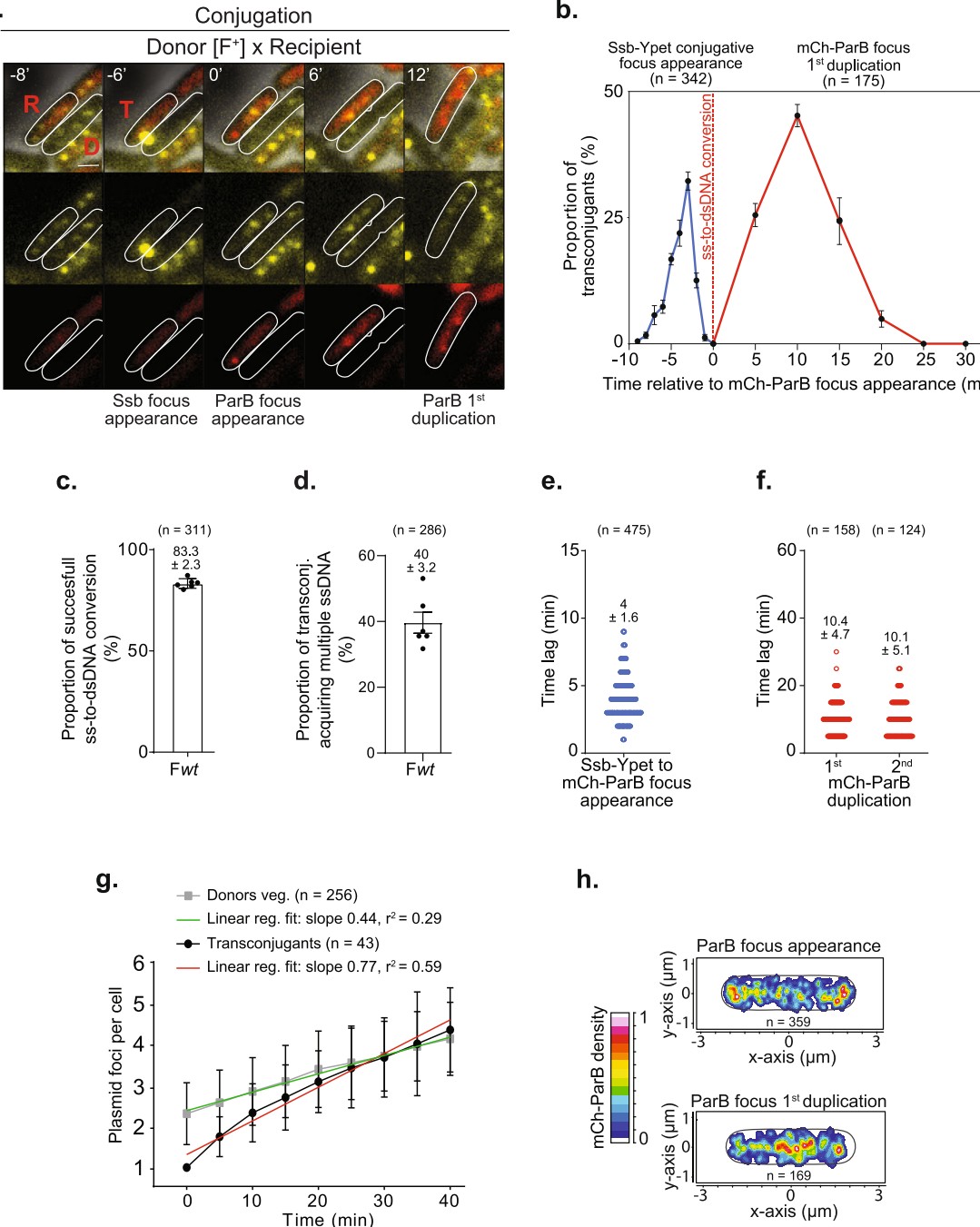

restored. This accelerated plasmid replication allows for the rapid increase in F copy number before the division of the transconjugant cells, thus facilitating the segregation of plasmid copies to daughter cells.

Localisation analysis reveals that the ss-to-dsDNA conversion and the first duplication event occur at distinct subcellular positions. The initial mCh-ParB focus preferentially appears in the polar region of the transconjugant cell, comparable to the ssDNA's entry location (compare Fig. 2h to Fig. 1c and Fig. S3b to Fig. S2a). A noticeable difference is that mCh-ParB foci appear less peripheral, indicating that they are not as close to the cell membrane as Ssb-Ypet conjugation foci (compare Fig. 2h to Fig. 1c and Fig. S3c to Fig. S2b). We observe that the mCh-ParB focus subsequently migrates to the midcell position before duplication (Fig. 2h and Fig. S3b, c). These data show that the two DNA synthesis reactions involved in plasmid processing (i.e. ss-to-dsDNA conversion and plasmid replication) are separated in time and space in

the new host cell. The recruitment of the complementary strand synthesis machinery and the ss-to-dsDNA replication reaction occurs in the vicinity of the polar position of entry of the ssDNA plasmid, while plasmid replication occurs in the midcell region. Altogether, these analyses reveal that plasmid processing steps (ssDNA entry, ss-to-dsDNA conversion and plasmid replication) occur at specific intracellular positions within the new host cell and follow a precise chronology.

## Programme of plasmid-encoded protein production in transconjugant cells

We constructed *superfolder gfp* (*sfgfp*) translational fusions to the 3′ end of several genes located in the different functional regions of the F plasmid to examine the production timing of plasmid-encoded proteins in transconjugant cells, which we use to get insights into the timing of plasmid gene expression (Fig. 3a and Fig. S4a). The *ygfA*,

**Fig. 2 | Timing and spatial localisation of the ss-to-dsDNA conversion and plasmid duplication in transconjugant cells. a** Time-lapse images showing ssDNA plasmid transfer reported by the formation of the Ssb-Ypet conjugative foci in both donor (D) and recipient (R) cells, followed by the ss-to-dsDNA conversion reflected by the appearance of an mCh-ParB focus in transconjugant (T) cells. Scale bar 1 μm. **b** Single-cell time-lapse quantification of Ssb-Ypet focus appearance (blue line) and mCh-ParB focus first duplication (red line) with respect to the ss-to-dsDNA conversion revealed by mCh-ParB focus formation in transconjugant cells (0 min). Ssb-Ypet focus appearance was analysed using 1 min/frame time-lapses, while mCh-ParB first and second duplication were analysed using 5 min/frame time-lapses. The mean and SD calculated from the indicated number of conjugation events analysed (*n*) from seven independent experiments is indicated. **c** Histogram of successful ss-to-dsDNA conversion reflected by the conversion of the Ssb-Ypet conjugative foci into an mCh-ParB focus. The mean and SD are calculated from (*n*) individual transfer events from six biological replicates (black dots). **d** Histogram showing the percentage of transconjugants with an mCh-ParB focus that acquires multiple ssDNA plasmids as revealed by the successive appearance of Ssb-Ypet conjugative focus. The mean and SD are calculated from (*n*) individual transconjugant cells from six biological replicates (black dots). **e** Scatter plot showing the time lag between Ssb-Ypet and mCh-ParB foci appearance in transconjugants. The mean and SD calculated from (*n*) individual events (blue circles) from seven biological replicates are indicated. **f** Scatter plot showing the time lag between the apparition of the mCh-ParB focus and its visual duplication in two foci (first duplication), and in three or four foci (second duplication). The mean and SD calculated from (*n*) individual duplication events (red circles) from at least six biological replicates are indicated. **g** Single-cell time-lapse quantification of the number of F foci per cell in F-carrying donor strain during vegetative growth and in transconjugants after F plasmid acquisition. For donors, the number of F foci per cell (number of SopB-sfGFP foci) with respect to cell birth (*t* = 0 min) is shown (grey curve). For transconjugants, the number of F foci per cell (number of mCh-ParB foci) with respect to mCh-ParB focus appearance (*t* = 0 min) is shown (black curve). Mean and SD calculated from (*n*) individual cells from four biological replicates are indicated, together with curves' linear fitting lines (green and red). F-carrying donor strain (LY834), Transconjugant (LY358 after F*wt* acquisition). **h** 2D localisation heatmaps of mCh-ParB foci at the time of its appearance (top) and just before its duplication (bottom). Heatmaps are normalisation by the cell length of (*n*) individual transconjugant cells from seven biological replicates. **a–f, h** F*wt* donor (LY1007), recipient (LY358) and transconjugant (LY358 after F*wt* acquisition). Source data are provided as a Source Data file.

*ygeA*, *psiB*, *yfjB*, *yfjA* and *ssb*^F genes are located in the leading region and are transferred in order after the origin of transfer *oriT*. The *sopB* gene is part of the SopABC partition system and is located in the maintenance region. The *traM*, *traC*, *traS* and *traT* genes are located in the *tra* region that encodes factors involved in plasmid transfer. TraM is the accessory protein of the relaxosome complex that is recruited to the *oriT*[65]; TraC is the traffic ATPase organised as a hexamer of dimers docked to the cytoplasmic faces of the T4SS[66]; TraS and TraT correspond to the F plasmid exclusion (immunity) system that protects against self-transfer[67–69].

We first performed time-course experiments where microscopy snapshot images of the conjugating population were acquired 1, 2, 4 and 6 h after mixing donors and recipient cells. For each time point, the frequency of transconjugants (T/R + T) was directly measured at the single-cell level from the proportion of recipient cells exhibiting diffuse mCh-ParB fluorescence (R) or transconjugant cells harbouring mCh-ParB foci (T), and the intracellular green fluorescence signal to noise ratio (SNR) was automatically measured (Fig. S4b–d). This snapshot analysis shows that all F plasmid derivatives carrying sfGFP fusions retained their transfer ability and yielded frequencies of transconjugants between 57 and 95% after 6 h of mating. Also, fusion-carrying plasmid acquisition is systematically followed by an increase in sfGFP signal in transconjugant cells, with highly variable timing and levels (Fig. S4b–d).

Better resolution of the production level and timing of sfGFP fusions with respect to the ss-to-dsDNA conversion (appearance of the mCh-ParB focus) in individual transconjugant cells was obtained using time-lapse imaging of conjugation performed in the microfluidic chamber (Movies S1, S2). We performed transconjugant cell detection and quantification of the intracellular sfGFP SNR cells over time (Fig. S5a–d). When the transconjugant cell divided, we continued fluorescence quantification in the resulting daughter cells to monitor sfGFP production over a longer period. From this raw data, we calculated the fold-increase in SNR per 10-min interval, where a fold-increase superior to one reveals that the fusions are being produced in the transconjugants (Fig. S5a–d). These data were finally translated into a comprehensive diagram presenting the production time windows for each fusion in transconjugant cells relative to the ss-to-dsDNA conversion event (Fig. 3b). This analysis reveals that fusions belonging to the different plasmid regions exhibit specific production timings with respect to plasmid processing steps.

Remarkably, we detect the synchronous production of the leading YgeA, PsiB, YfjB, YfjA and Ssb^F fusion proteins even before the appearance of the mCh-ParB focus (Fig. 3b and Fig. S5a). Furthermore, the production of these fusions is only transient as it peaks at -5 min and stops 25–35 min after the ss-to-dsDNA conversion event. This unexpected observation indicates that leading fusions start being produced when the plasmid is still in ssDNA form and stops rapidly after the plasmid is converted into dsDNA form. An interesting exception is YgfA-sfGFP, for which production is only detected in the 10–20 min interval after mCh-ParB focus appearance. The *ygfA* gene is the closest to the *oriT* and is, therefore, the first gene to be transferred to the recipient (Fig. 3a and Fig. S4a). However, *ygfA* gene orientation is opposite to other tested leading genes, meaning that the T-strand does not correspond to the template strand for *ygfA* transcription. Consequently, and consistent with our observations, *ygfA* expression can only occur after synthesising the complementary template strand by the ss-to-dsDNA conversion.

The ss-to-dsDNA conversion is followed by the production of maintenance and Tra proteins, starting with SopB and TraM, then TraC, and eventually TraS and TraT fusions (Fig. 3b and Fig. S5b, c). The production of these fusions is expected to require the presence of the plasmid in dsDNA form since the corresponding genes are known to be controlled by dsDNA promoters ($P_{sopAB}$ for *sopB*, $P_M$ for *traM* and $P_Y$ for *traC* and *traST*). However, what could explain the observed differences in the production timings? We addressed whether timing discrepancies could simply account for the fusions' position on the genetic map of the F plasmid. This possibility was excluded by the observation that insertion of the constitutive fluorescent reporter $P_{lacIQ1}$sfGFP (*sfgfp* gene under the control of the $P_{lacIQ1}$ constitutive promoter) in the *repE-sopA*, *tnpA-ybaA* and *traM-traJ* intergenic regions resulted in similar sfGFP production timings, within the 0–10 min interval after the appearance of the mCh-ParB focus (Fig. 3b and Fig. S5d). Instead, we propose that the differential production timings of maintenance and *tra* genes reflect the activity and regulation of the promoters of the corresponding genes. The *sopAB* operon is under the control of the $P_{sopAB}$ promoter, which is repressed by SopA binding. Therefore, the $P_{sopAB}$ promoter is expected to be fully unrepressed and active in transconjugant cells devoid of SopA, thus allowing the rapid production of the SopAB partition complex required for plasmid stability and inheritance over cell divisions. The *traM* gene is controlled by the $P_M$ promoter, which is weakly but constitutively active, even before its full activation by binding the TraY protein[70]. By contrast, the $P_Y$ promoter that controls the expression of *traC*, *traS* and *traT* genes needs to be activated by the TraJ protein, encoded by the *traJ* gene under the control of its own promoter $P_J$ and located upstream of $P_Y$[4]. The requirement for this activation cascade probably explains

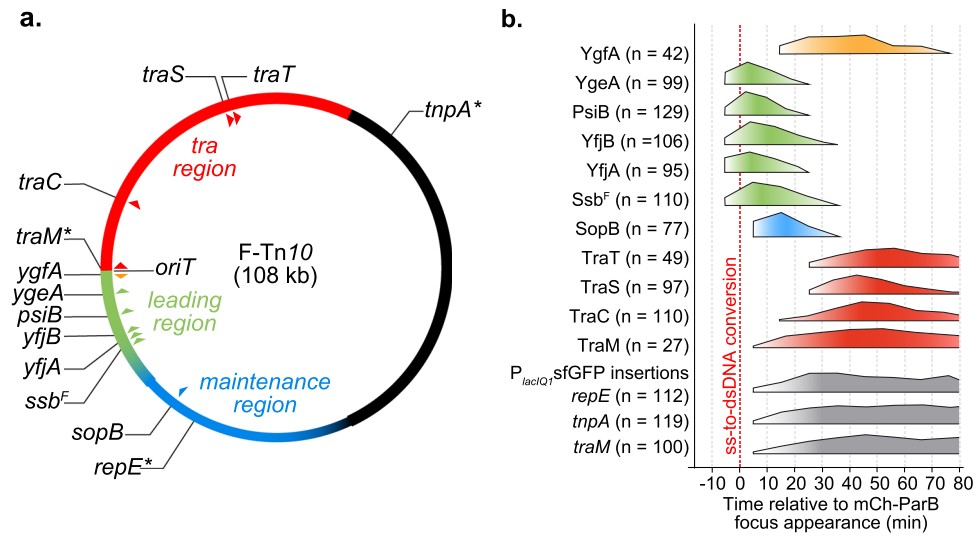

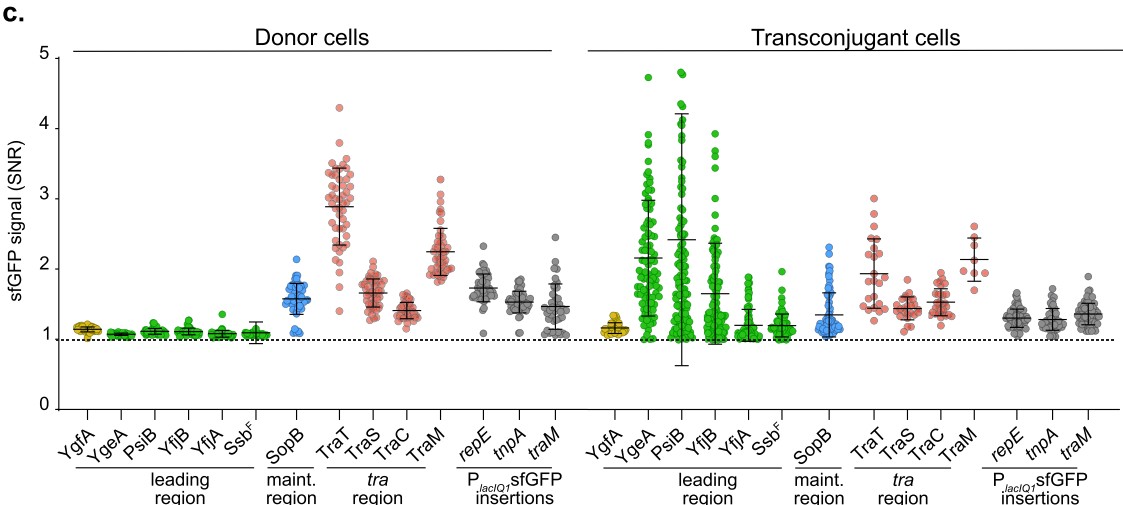

**Fig. 3 | Timing of plasmid-encoded proteins production in transconjugant cells. a** Genetic map of the 108 kb F plasmid indicating the leading (green), *Tra* (red) and maintenance (blue) regions, and the positions of the studied genes (triangles). Stars represent the genetic location of the P*lacIQ1sfgfp* insertions. **b** Summary diagram of the production timing of each plasmid-encoded protein fusions in transconjugant cells with respect to the timing of ss-to-dsDNA conversion reflected by mCh-ParB focus appearance (0 min). The diagram represents data from the foldchange increase in sfGFP signal from Fig. S5. Orange/green, blue and red colours correspond to production of proteins from the leading, maintenance and transfer region, respectively. Timings of the cytoplasmic sfGFP production from the P*lacIQ1* promoter inserted in the *repE-sopA* (*repE*), *tnpA-ybaA* (*tnpA*) and *traM-traJ* (*traM*) intergenic regions are represented in grey. The number (*n*) of individual transconjugant cells from at least three biological replicates analysed is indicated. **c** Jitter plots showing the intracellular green fluorescence (SNR) for each sfGFP fusions and reporters within vegetatively growing donor (left) and transconjugant cells (right) at the maximum SNR value from Fig. S5. Each dot represents data of individual cells. Means and SD are calculated from the indicated (*n*=) number of transconjugant cells from at least three independent biological replicates. Donors of F derivatives (see Table S1), Recipient (LY358). Source data are provided as a Source Data file.

the delayed production of TraC, TraS and TraT. The additional delay between TraC and TraS/TraT fusions production could potentially reflect the relative distance of these genes to the $P_Y$ promoter (5.9 kb for *traC* and 20.4 kb for *traST*). It is important to stress that the F plasmid carries a naturally occurring insertion of the IS3 insertion sequence into the *finO* gene of the FinOP fertility inhibition system, which results in the upregulation and constitutive expression of *tra* genes[71]. Therefore, other IncF plasmids in which the FinOP regulatory system is still active are expected to exhibit different timings and production levels of the Tra proteins than reported here.

Notably, the intracellular levels of Tra proteins within transconjugant cells reach a plateau between 60 to 90 min after the ss-to-dsDNA conversion and remain stable throughout our observations

(Fig. 3b and Fig. S5c). This involves that, at that point, transconjugant cells have produced the transfer machinery and the exclusion system and have most likely been converted into proficient plasmid donors. In support of this interpretation, TraM, TraC, TraS, TraT and SopB are detected at similar levels in vegetatively growing F-carrying donor cells (Fig. 3c and Figs. S4c, d, S5b, c). This is not the case for YgeA, PsiB, YfjB, YfjA and Ssb[F] leading proteins, which intracellular levels start decreasing 25–35 min after the ss-to-dsDNA conversion in the transconjugants, and which are not detected in vegetatively growing donor cells (Fig. 3c and Figs. S4b, S5a). These results are consistent with the interpretation that leading proteins are produced rapidly and only transiently upon entry of the ssDNA plasmid in the recipient cells and not when the plasmid is maintained in dsDNA form during vegetative replication.

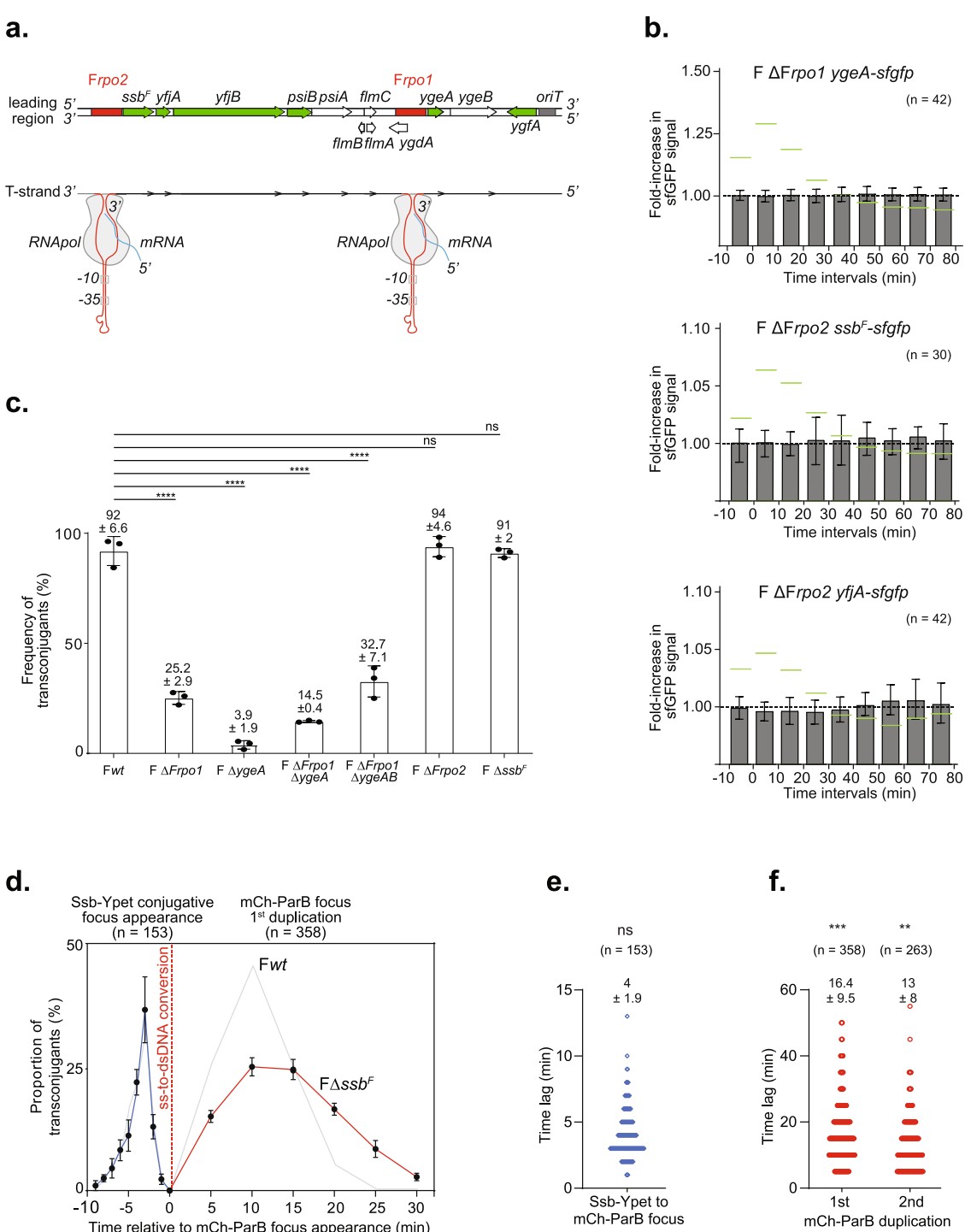

## Single-stranded promoters allow the early expression of the leading genes in the transconjugant cell

Together with previous works[37,38,54,56], the early and transient expression of leading genes in transconjugant cells supports the existence of specific sequences that would act as single-stranded promoters to initiate the transcription of leading genes from the internalised ssDNA plasmid. Using bioinformatics analysis, we identified a region upstream of the *ssb[F]*, *yfjA*, *yfjB*, *psiA* and *psiB* genes, which we named F*rpo2*, that shares 92% identity with the previously reported F*rpo* region (renamed F*rpo1*) located upstream *ygeA* and *ygeB* and previously characterised in vitro[55] (Fig. 4a). DNA folding prediction using mFold (http://www.unafold.org) indicates that the single-stranded form of F*rpo2* can fold into a highly stable stem-loop structure that also

carries canonical −10 and −35 boxes, similar to the F*rpo1* region (Fig. S6a)[55]. We addressed the effect of F*rpo1* or F*rpo2* deletions on the expression of the downstream genes in transconjugant cells using live-cell microscopy. Microscopy analysis of transconjugant cells receiving the F ΔF*rpo1 ygeA-sfgfp*, the F ΔF*rpo2 ssb[F]-sfgfp*, or the F ΔF*rpo2 yjfA-sfgfp* revealed no significant fold-increase in sfGFP fluorescence before or after the ss-to-dsDNA conversion in the transconjugant cells (Fig. 4b).

We then addressed the impact of F*rpo1* and F*rpo2* deletions on the efficiency of conjugation after three hours of mating, as estimated by plating assays (Fig. 4c). F ΔF*rpo1* exhibits a significantly reduced frequency of transconjugants of 25.2 ± 2.9% compared to 92.6 ± 6.6% for the F*wt*. Comparable results were obtained for F ΔF*rpo1* Δ*ygeAB*

**Fig. 4 | Role of leading region factors Frpo1, Frpo2 and ssb$^F$ in conjugation.**
**a** Genetic map of the leading region showing the position of the genes (green for studied sfGFP fusions and white for the other genes) and Frpo1 and Frpo2 promoters (red) (top). The bottom diagram shows the stem-loop structure formed by the ssDNA forms of Frpo1 and Frpo2 sequences (detailed in Fig. S6). **b** Histograms of intracellular sfGFP fold-increase in transconjugants after the acquisition of F ΔFrpo1 ygeA-sfgfp, F ΔFrpo2 ssb-sfgfp and F ΔFrpo2 yfjA-sfgfp. Mean and SD are calculated from the indicated (n) individual transconjugant cells analysed from at least three independent experiments. Levels obtained with the Fwt plasmid from Fig. S5a are wt reported in green as a reference. Donor of F ΔFrpo1 ygeA-sfgfp (LY1368), F ΔFrpo2 ssb-sfgfp (LY1365), F ΔFrpo2 yfjA-sfgfp (LY1364) and the recipient (LY318). **c** Histograms of Fwt, deletion mutants F ΔFrpo1, F ΔygeA, F ΔFrpo1 ΔygeA, F ΔFrpo1 ΔygeAB, FΔFrpo2 and FΔssb$^F$ frequency of transconjugant (T/R + T) estimated by plating assays. Mean and SD are calculated from three independent experiments (shown as individual black dots). P value significance ns and ****P ≤ 0.0001 were obtained from one-way ANOVA with Dunnett's multiple comparisons test. Donor of Fwt (LY875), F ΔFrpo1 (LY824), F ΔygeA (LY160), F ΔFrpo1 ΔygeA (LY1424), F ΔFrpo1 ΔygeAB (LY1425), F ΔFrpo2 (LY823), F Δssb$^F$ (LY755), recipient (MS428). **d** Single-cell time-lapse quantification of Ssb-Ypet focus appearance (blue line) and mCh-ParB focus first duplication (red line) with respect to mCh-ParB focus formation in transconjugant cells (0 min) that receive the FΔssb$^F$ plasmid. The number of conjugation events analysed (n) from five independent biological replicates is indicated. Results obtained in Fig. 2b with Fwt plasmid are reported in grey for comparison. **e** Scatter plot showing the time lag between the appearance of the Ssb-Ypet focus and the appearance of the mCh-ParB focus in transconjugant cells after the acquisition of the F Δssb$^F$ plasmid. The mean and SD calculated from (n) individual ss-to-dsDNA conversion event (blue circles) from five biological replicates are indicated. P value significance ns (>0.05 non-significant) was obtained from Mann–Whitney two-sided statistical test against results obtained with the Fwt plasmid (Fig. 2e). **f** Scatter plot showing the time lag between the apparition of the mCh-ParB focus and its visual duplication in two foci (first duplication), and in three or four foci (second duplication) in transconjugant cells after acquisition of the F Δssb$^F$ plasmid. The mean and SD calculated from (n) individual duplication events (red circles) from eight biological replicates are indicated. P value significance **P = 0.0023 and ***P = 0.0007 were obtained from Mann–Whitney two-sided statistical test against results obtained with the Fwt plasmid (Fig. 2f). Donor F Δssb$^F$ (LY1068), recipient (LY358). Source data are provided as a Source Data file.

(32.7 ± 7.1) and F ΔFrpo1 ΔygeA (14.5 ± 0.4). Surprisingly, the single deletion of ygeA decreases the conjugation efficiency even further (3.9 ± 1.9%), and despite our multiple attempts, the deletion of ygeB alone could never be constructed. By contrast, the deletions of Frpo2 or ssb$^F$ have no significant impact on the conjugation efficiency. These results show that Frpo1 and Frpo2 are required for the early expression of the downstream genes upon plasmid entry in recipient cells during conjugation in vivo. However, genes under the control of Frpo1 appear to have a more critical role in conjugation than those under the control of Frpo2.

### Role of the plasmid-encoded Ssb$^F$ leading protein in plasmid establishment

The rapid and transient expression of leading genes upon plasmid entry strongly suggests that leading proteins have an essential role during the early steps of plasmid establishment in the new host cell. The leading region conserved in various enterobacterial plasmids encodes a homologue of the single-strand-binding protein Ssb encoded on the *E. coli* chromosome[41,43,54,72–74]. The chromosomally encoded ssb gene is conserved and essential in all bacterial organisms, raising the question of the *raison d»être* of plasmid-born ssb homologues. Early study shows that the Ssb$^F$ encoded by the F plasmid can partially complement conditional mutations of the chromosomal ssb gene[73,75]. Consistently, we performed simultaneous visualisation of Ssb$^F$-mCh produced from a pTrc99a-ssb$^F$-mch plasmid and the chromosomally encoded Ssb-Ypet (Fig. S7a) and observed similar intracellular positioning (Fig. S7b) confirmed by colocalisation analysis (Fig. S7c). This indicates that both the plasmid Ssb$^F$ and the host Ssb are recruited to the ssDNA that follows the replication forks in vegetatively growing cells. Similarly, Ssb$^F$-sfGFP also forms foci in transconjugant cells that have acquired the F ssb$^F$-sfgfp plasmid, mainly during the first and second plasmid duplication events (Fig. S7d, e). Nonetheless, the role of Ssb$^F$ during conjugation is still unclear, and its deletion from the F plasmid has no significant impact on conjugation efficiency (Fig. 4c).

To get further insight into the role of Ssb$^F$ during conjugation, we revisited the dynamics of ssDNA entry, ss-to-dsDNA conversion and duplication of the F Δssb$^F$ plasmid. Time-lapse microscopy image analysis reveals that Ssb$^F$ deletion has no impact on the dynamics of Ssb-Ypet conjugative foci (Fig. 4d) or the timing of the ss-to-dsDNA conversion (compare Fig. 4e to Fig. 2e). However, Ssb$^F$ deletion dramatically delays the timing of plasmid duplication in transconjugant cells (compare Fig. 4f to Fig. 2f). The time lag between mCh-ParB appearance and the first duplication is increased by ~58% (from 10.4 ± 4.7 for Fwt to 16.4 ± 9.5 for F Δssb$^F$), and the time between the first and second plasmid replication event is increased by ~29% (from 10.1 ± 4.7 for Fwt to 13 ± 8 for F Δssb$^F$). This indicates that Ssb$^F$ has a role in facilitating the first rounds of plasmid duplication in the new transconjugant cell, possibly by increasing the cellular pool of single-strand binding protein available for DNA replication. This function appears dispensable since the absence of Ssb$^F$ delays plasmid duplication but does not affect the final efficiency of conjugation, at least when conjugation is performed in optimal conditions between *E. coli* MG1655 strains.

## Discussion

Our current knowledge of conjugation mainly emerges from experimental genetic, biochemical and structural studies that provided a well-documented understanding of the molecular reactions and factors involved in DNA transfer, while genomic and computational studies uncovered the diversity of conjugative plasmids and their importance in the epidemiology of antibiotics resistance dissemination. It is only recently that the application of optical microscopy has started to provide insights into the organisation of conjugation at the cellular scale[58,59,76–82]. In this study, live-cell microscopy combined with specifically developed fluorescent reporters offers a unique view of the cellular dynamics of conjugation while providing insights into the timing and localisation of each key step.

We report the presence of ssDNA plasmid on both the donor's and the recipient's side during plasmid transfer. Noticeably, the ssDNA plasmid is not randomly positioned but instead allocated to specific subcellular locations within the mating pair cells. The exit point of the ssDNA F plasmid is preferentially located on the side of the donor cell and preferentially at quarter positions. This pattern could reflect the intracellular position of the T4SS machinery of the F plasmid, which to our knowledge, remains to be described. This possibility would be weakened if the F plasmid T4SS machinery is homogeneously located throughout the periphery of the cells as in the case of the pTi and R388 plasmids[79,81]. Alternatively, the lateral localisation of active conjugation pores may reflect the facilitated access to F plasmid molecules, which are also positioned at quarter positions and excluded from the cell poles[83,84]. By contrast, the ssDNA mainly enters the polar region of the recipient cells. This could suggest that the pole of the recipients' surface is the preferred location for the donor's F pilus. attachment or the stabilisation of the mating pair. The latter possibility is reinforced by the fact that mating pair stabilisation during F conjugation involves interaction between the plasmid protein TraN exposed at the surface of the donor cells and the host outer membrane protein OmpA of the recipient cells[82,85]. OmpA was shown to be enriched and less mobile in the polar regions of *E. coli* cells[86], possibly favouring the stabilisation of the mating pair and the conjugation pore at this location.

a.

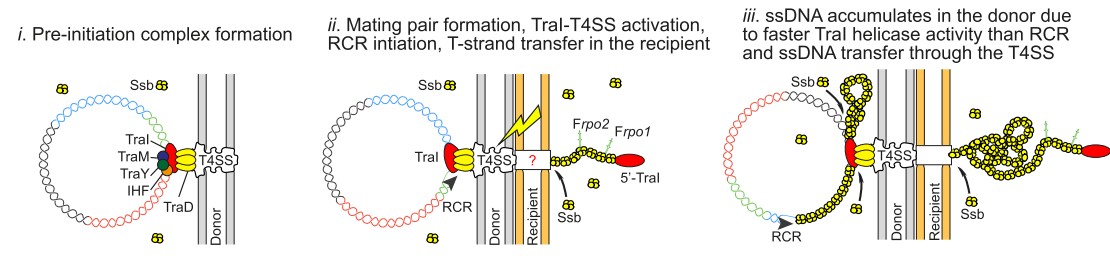

b.

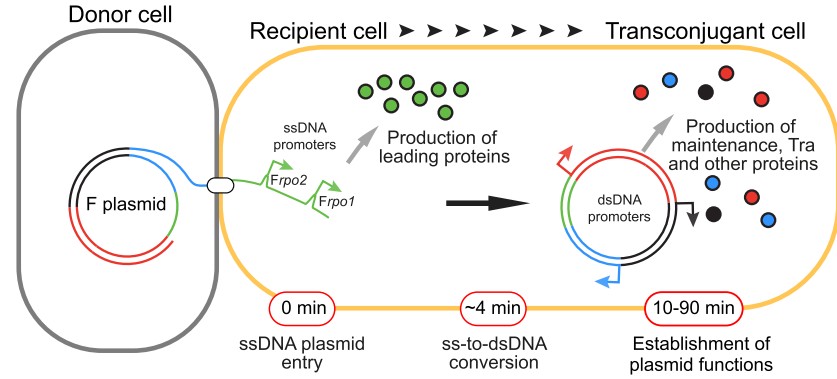

**Fig. 5 | Model for conjugation initiation and intracellular dynamics. a**(i) Before the initiation of conjugation, the pre-initiation complex bound to the plasmid's origin of transfer is docked to the Type IV secretion system (T4SS). (ii) The establishment of the mating pair transduces a signal that activates the pre-initiation complex. The unwinding of the dsDNA plasmid by the helicase activity of TraI produces the first segment of the T-strand, which is immediately transferred into the recipient cell where it recruits Ssb molecules, while the non-transferred strand is complemented by rolling-circle replication (RCR) in the donor cell. (iii) The helicase activity of TraI generates ssDNA at a higher rate than the T-strand is transferred through the T4SS or the non-transferred strand is complemented by

RCR, thus resulting in the accumulation of ssDNA plasmid coated by Ssb molecules in the donor cell. **b** Upon entry of the ssDNA plasmid in the recipient cell, F*rpo1* and F*rpo2* leading sequences form stem-loop structures that serve as promoters initiating the transcription of the downstream leading genes, rapidly resulting in the production of leading proteins. The subsequent ss-to-dsDNA conversion inactivates F*rpo1* and F*rpo2* and licences the expression of other plasmid genes under the control of conventional dsDNA promoters. The production of maintenance, transfer and other plasmid-encoded proteins eventually results in the development of new functions by the transconjugant cell.

The unexpected finding that the ssDNA first appears in the recipient cell, and later accumulates in the donor during conjugation also provides insights into the activity of TraI and its coordination with the transfer of the T-strand through the T4SS or the RCR of the non-transferred strand. Before DNA transfer initiation, the relaxosome bound to the plasmid's *oriT* is docked to the T4SS by the TraD coupling protein, thus forming the pre-initiation complex (Fig. 5a(i)). Contact with the recipient cell is proposed to induce a signal that activates the pre-initiation complex. We uncover the existence of a brief period where part of the T-strand has already been transferred into the recipient cell while no ssDNA is present within the donor (Fig. 5a(ii)). At this stage, the absence of ssDNA in the donor implicates that all the ssDNA generated by TraI has been removed, both by transfer of the T-strand through the T4SS and by complementation of the non-transferred ssDNA strand by the RCR. After this transient step, the ssDNA also accumulates in the donor, suggesting that the ssDNA is generated by TraI helicase activity in the donor faster than it is removed by transfer and RCR synthesis (Fig. 5a(iii)).

Assuming the 2.9 ± 1.1 min lifespan of the Ssb-Ypet foci in transconjugants reflects the time required to complete the internalisation of the 108,000 nt ssDNA F plasmid, we calculated a $620 \pm 164$ nt s$^{-1}$ transfer rate. This is in reasonable agreement with the historical 770 nt s$^{-1}$ rate estimated from the 100 min required to transfer the whole 4.6 Mb *E. coli* chromosome[87]. Besides, the rate of DNA synthesis by the DNA polymerase III holoenzyme during RCR was estimated at 650–750

nuc s$^{-1}$ [88]. By comparison, the rate of TraI helicase activity was measured at $1120 \pm 160$ bp s$^{-1}$[89]. These estimates support the view that ssDNA accumulation in the donor accounts for the faster rate of TraI helicase activity than the rate of T-strand plasmid transfer or RCR. Therefore, it is possible that, contrasting with the previously suggested but never demonstrated proposal, the helicase activity of the relaxase is not strictly coupled with the activity of DNA translocation through the T4SS.

Live-cell microscopy uncovers the global chronology conjugation steps, as summarised in Fig. 5b. The plasmid processing in the transconjugant cell is a relatively rapid process, as the entry of the ssDNA plasmid and its conversion into dsDNA is completed in about 4 min on average. Most importantly, the ss-to-dsDNA conversion event is the pivotal event that determines the programme of plasmid gene expression. Leading genes are the first to enter the recipient cell and also the first to be expressed from the F plasmid in ssDNA form. Consistently with previous proposals[55–57], we show that the early expression of leading genes depends on sequences that act as single-stranded promoters when the plasmid is still in ssDNA form. As previously described for F*rpo1*, we propose that the highly homologous F*rpo2* sequences identified here folds into a stable stem-loop structures that reconstruct −35 and −10 consensus boxes, resulting in transcription initiation.

Leading gene expression is also transient as the ss-to-dsDNA conversion turns off leading protein production by inactivating F*rpo1*

and Frpo2 promoters while licencing the expression of maintenance, transfer and other plasmid genes under the control of conventional dsDNA promoters, often subject to their own regulation specificities. Maintenance and transfer protein levels within transconjugants reach a steady-state equivalent to that of vegetatively growing F-containing cells in about 30 to 90 min, depending on the protein. Interestingly, our previous work showed that tetracycline resistance factors encoded by the Tn10 transposon inserted in the intergenic region ybdB-ybfA of the F plasmid are also produced immediately after the ss-to-dsDNA conversion and reach the resistant cell's level within ~90 min[58]. These findings consistently indicate that this time scale corresponds to the period needed for the transconjugant cells to gain plasmid-encoded functions, including plasmid maintenance, conjugation ability, immunity against self-transfer and additional resistance potentially carried by the plasmid.

The regulation of plasmid gene expression by plasmid processing is an elegant way to ensure the sequential and timely production of plasmid proteins in the transconjugant cell, and particularly to restrict the production of leading factors to a narrow time window following the entry of the ssDNA plasmid. However, de novo protein synthesis might not be the only way to provide the transconjugant cell with plasmid-encoded proteins. Recent work by Al Mamun et al. reports that the transfer of the F-like plasmid pED208 (IncFV) is concomitant with the translocation of several plasmid-encoded proteins, including TraI, ParA, ParB1, Ssb homologue Ssb^ED208, ParB2, PsiB and PsiA[90]. Protein translocation was detected at low frequency ($10^{-5}$ recombinants per donor cell between one and five hours of mating) using a highly sensitive Cre recombinase assay. Protein translocation might also occur during the transfer of the native F plasmid but could not solely explain our observations. Indeed, our microscopy analysis shows that YgeA, PsiB, YfjB, YfjA and Ssb^F leading fusions are below the microscopy detection threshold in donor cells but are quantified at significant intracellular levels in all transconjugant cells. This implies that the amounts of leading proteins observed in the transconjugant cells cannot just originate from donor cells, but result from de novo protein synthesis, which we show depends on Frpo1 and Frpo2 sequences. Therefore, it appears likely that direct protein translocation and de novo synthesis are concomitant mechanisms ensuring the presence of leading factors and associated functions immediately upon entry of the ssDNA plasmid in the transconjugant cell. This further suggests the critical role of the leading region in conjugation. Several elements support this view. The leading region is conserved in a variety of conjugative plasmids[41–46]. In addition, the leading regions of plasmids belonging to a wide range of incompatibility groups (IncF, IncN, IncP9 and IncW) classified as MOBF plasmids using the relaxase as a phylogenetic marker were reported to be the preferential target for CRISPR-Cas systems directed against conjugation[8,91,92]. Recently, the leading region was shown to be an important evolutionary target for the dissemination of the pESBL (IncI) plasmid[93]. Concerning the F plasmid, we can stress that Frpo1 and Frpo2 share 92% similarity at the nucleotide level and are located only about 5 kb apart. This implies that when in dsDNA form during vegetative plasmid replication, Frpo1 and Frpo2 sequences would be a potential substrate for homologous recombination, resulting in the deletion of the intervening segment. However, the intervening segment carries the flmAB genes, functional homologues to the hok/sok toxin-antitoxin system from the R1 plasmid[94], which are likely to safeguard the stability of the leading region.

Despite this body of evidence, it is currently challenging to rationalise the importance of the leading region since the molecular functions of most leading proteins are still unknown. Our data indicate that genes downstream of Frpo1 (ygeA et ygeB) have a critical function in conjugation. By contrast, genes located downstream Frpo2 (ssb^F, yfjA, yfjB, psiB, psiA and flmC) appear to be dispensable since deletions of Frpo2, ssb^F or psiB[44] have no significant impact on the overall conjugation efficiency addressed by plating assays. Yet, Ssb^F and PsiB have been shown to suppress conjugation-induced SOS induction in the transconjugant cell[43,54,90], which is likely important for the transconjugant's physiology and proliferation rather than for plasmid transfer per se. One potential limitation of the conjugation study is that transfer efficiency assays are generally performed between identical or closely related bacterial strains in optimal medium and temperature conditions. This likely undermines the role of genes that are not strictly essential but might facilitate or optimise conjugation, or help the proliferation of the transconjugant cell. Hence, it is possible that the importance of the leading factors would be best revealed in less favourable conditions, between phylogenetically distant bacteria, or on the evolutionary scale. Meanwhile, real-time microscopy might help uncover the potentially subtle influence of these genes on the sequence of conjugation in live cells.

## Methods

### Bacterial strains, plasmids and growth
Bacterial strains are listed in Table S1, plasmids in Table S2 and oligonucleotides in Table S3. Fusion of genes with fluorescent tags and gene deletion on the F plasmid used λRed recombination[95,96]. Modified F plasmids were transferred to the background strain K12 MG1655 by conjugation. Where multiple genetic modifications on the F plasmid were required, the kan and cat genes were removed using site-specific recombination induced by expression of the Flp recombinase from plasmid pCP20[95]. Plasmid cloning were done by Gibson Assembly and verified by Sanger sequencing (Eurofins Genomics biotech). Strains and plasmids were verified by Sanger sequencing (Eurofins Genomics). Cells were grown at 37 °C in an M9 medium supplemented with glucose (0.2%) and casamino acid (0.4%) (M9-CASA) before imaging, and in Luria-Bertani (LB) broth for conjugation efficiency assays. When appropriate, supplements were used in the following concentrations; Ampicillin (Ap) 100 μg/ml, Chloramphenicol (Cm) 20 μg/ml, Kanamycin (Kn) 50 μg/ml, Streptomycin (St) 20 μg/ml and Tetracycline (Tc) 10 μg/ml.

### Conjugation assays
Overnight cultures in LB of recipient and donor cells were diluted to an $A_{600}$ of 0.05 and grown until an $A_{600}$ comprised between 0.7 and 0.9 was reached. 25 μl of donor and 75 μl of recipient cultures were mixed into an Eppendorf tube and incubated for 90 min at 37 °C. About 1 ml of LB was added gently and the tubes were incubated again for 90 min at 37 °C. The conjugation mix were vortexed, serial diluted, and plated on LB agar X-gal 40 μg/ml IPTG 20 μM supplemented with the appropriate antibiotic to select for recipient or donor populations. Recipient (R) colonies were then streaked on plated on LB agar containing tetracycline 10 μg/ml to select for transconjugants (T) and the frequency of transconjugant calculated from the (T/R + T) presented in Fig. 4c.

### Live-cell microscopy experiments
Overnight cultures in M9-CASA were diluted to an $A_{600}$ of 0.05 and grown until $A_{600} = 0.8$ was reached. Conjugation samples were obtained by mixing 25 μl of the donor and 75 μl of the recipient into an Eppendorf tube. For time-lapse experiments, 50 μl of the pure culture or conjugation mix was loaded into a B04A microfluidic chamber (ONIX, CellASIC®)[97]. The nutrient supply was maintained at 1 psi and the temperature was maintained at 37 °C throughout the imaging process. Cells were imaged every 1 or 5 min for 90 to 120 min. For snapshot imaging, 10 μl samples of clonal culture or conjugation mix were spotted onto an M9-CASA 1% agarose pad on a slide[98] and imaged directly.

**Image acquisition.** Conventional wide-field fluorescence microscopy imaging was carried out on an Eclipse Ti2-E microscope (Nikon), equipped with x100/1.45 oil Plan Apo Lambda phase objective, ORCA-

Fusion digital CMOS camera (Hamamatsu), and using NIS software for image acquisition. Acquisitions were performed using 50% power of a Fluo LED Spectra X light source at 488 and 560 nm excitation wavelengths. Exposure settings were 100 ms for Ypet, sfGFP and mCherry and 50 ms for phase contrast.

**Image analysis.** Quantitative image analysis was done using Fiji software with MicrobeJ plugin[99]. For snapshot analysis, cells' outline detection was performed automatically using MicrobeJ and verified using the Manual-editing interface. For time-lapse experiments, detection of cells was done semi-automatedly using the Manual-editing interface, which allows to select the cells to be monitored and automatically detect the cell outlines. Within conjugation populations, donor (no mCh-ParB signal), recipient (diffuse mCh-ParB signal) or transconjugant (mCh-ParB foci) categories were assigned using the 'Type' option of MicrobeJ. Recipient cells were detected on the basis of the presence of red fluorescence above the cell's autofluorescence background level detected in the donors. Among these recipient cells, transconjugants were identified by running MicrobeJ automated detection of the ParB fluorescence foci (Maxima detection). This approach was used independently of the presence or the absence of the Ssb-Ypet, or sfGFP fusions within donor and recipient cells. Within the different cell types, mean intensity fluorescence (a.u.), skewness, signal/noise ratio (SNR) or cell length (μm) parameters were automatically extracted and plotted using MicrobeJ. SNR corresponds to the ratio (mean intracellular signal/mean noise signal), where the mean intracellular signal is the fluorescence signal per cell area and the noise is the signal measured outside the cells (due to the fluorescence emitted by the surrounding medium). By contrast with the total amount of fluorescence per cell, which is depending on the cell size/age and accounts for the background, SNR quantitative estimate is more appropriate for unbiased quantification of intracellular fluorescence over time. Ssb-Ypet, Ssb$^F$-mCh and mCh-ParB foci were detected using MicrobeJ Maxima detection function, and foci localisation and fluorescence intensity were extracted and plotted automatically. Plots presenting time-lapse data were either aligned to the first frame where the transconjugant cell exhibits a conjugative Ssb-Ypet focus (ssDNA acquisition) or an mCh-ParB focus (ss-to-dsDNA conversion) as indicated in the corresponding figure legend. Importantly, because conjugation is asynchronous in the population, time-lapse movies do not always capture the entire sequence of DNA transfer, i.e., Ssb-Ypet focus appearance/disappearance, mCh-ParB focus formation, first and second mCh-ParB duplication events. Of note, using 1 min/frame time-lapses was suitable to analyse Ssb-Ypet appearance/disappearance relative to the mCh-ParB formation (Fig. 2b–e), but provoked mCh-ParB bleaching thus hindering the analysis of mCh-ParB duplication events on the long term. We have then used 5 min/frame time-lapses to analyse mCh-ParB first and second duplication events relative to mCh-ParB focus formation (Fig. 2b, f). Also, the characterisation of the different transfer parameters was performed using specific analysis. For instance, time lags were calculated by counting the number of frames between the two considered events (Ssb-Ypet appearance and disappearance in Figs. 2e, 4e; mCh-ParB focus formation and first or second duplications in Figs. 2f, 4f). By contrast, Figs. 2b, 4b were generated by annotating the presence/absence of Ssb-Ypet or mCh-ParB foci in each time-lapse frame.

**Statistical analysis**
*P* value significance were analysed by running specific statistical tests on the GraphPad Prism software. Single-cell data from quantitative microscopy analysis were extracted from the MicrobeJ interface and transferred to GraphPad. *P* value significance of single-cell quantitative data was performed using unpaired non-parametric Mann–Whitney two-sided statistical test, which allows to compare differences between independent data groups without normal distribution

assumption. *P* value significance for the frequency of transconjugants obtained by plating assays were evaluated using One-way analysis of variance (ANOVA) with Dunnett's multiple comparisons test, which allows to determine the statistical significance of differences observed between the means of three or more independent experimental groups against a control group mean (corresponding to the F*wt*). When required, *P* value and significance are indicated on the figure panels and within the corresponding legend.

**Reporting summary**
Further information on research design is available in the Nature Portfolio Reporting Summary linked to this article.

## Data availability
All data to understand and assess the conclusions of this research are available in the main text and Supplementary Information. Source data are provided with this paper in the Source Data file. Raw microscopy data are available on Figshare (https://doi.org/10.6084/m9.figshare.21206444). Source data are provided with this paper.

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

## Acknowledgements

The authors thank the National BioResource Project and Coli Genetic Stock Center for providing strains, A. Ducret for valuable help with

MicrobeJ and N. Fraikin for helpful discussion. This research was funded by the Foundation for Medical Research, grant number FRM-EQU202103012587 to C.L. and A.C.; the French National Research Agency, grant number ANR-18-CE35-0008 to C.L., Y.Y., and K. G.; and the University of Lyon through funding to C.V. C.L. also acknowledges the Schlumberger Foundation for Education and Research (FSER 2019).

## Author contributions

C.L. and S.B. conceived, designed and supervised the execution of the study; A.C., C.V., K.G., A.B.-D., A.R., S.N. and S.B. performed the experiments and analysed the data. C.L. and S.B. wrote the paper and C.L. prepared the figures. C.L. and Y.Y. provided funding.

## Competing interests

The authors declare no competing interests.
