## [Peer Review File · Nature Communications]

Real-time visualisation of the intracellular dynamics of conjugative plasmid transferReviewer #1 (Remarks to the Author):

The study by Couturier et al. develops a single cell approach to examine the timing and the subcellular location of key steps of the conjugation process. Overall, the manuscript provides significant insights into the fundamental field of bacterial conjugation and open new perspectives to study this mechanism.

The authors use different fluorescent tools to monitor the transfer of the ssDNA plasmid from the donor to the recipient cells, and to detect its conversion into dsDNA. They first show that plasmid ssDNA accumulates both in the recipient and donor cells and propose that the helicase unwinding activity of the dsDNA plasmid is faster than its replication by rolling circle mechanism. They demonstrate that multiple transfers occur within an established mating pair and that the ss-to-dsDNA conversion is spatially separated from plasmid duplication. Finally, they measured the production time window and the level of proteins synthesized by the plasmid in transconjugant cells. Their analysis provides evidence that the differences of production timing rely on the specificities of the promoters controlling the genes encoding these proteins.

The paper is well written, the data is assembled in a clear and concise way and conclusions in general are elegantly supported by experimental results and statistical analyses.

Minor comments:

1. Line 133, Fig S2A: It is written that there is a noticeable enrichment of the Ssb-Ypet foci at the cell quarter positions in the donor cells. The histograms shown in Fig S2A indicate however that the proportion of foci located at the quarter positions is similar in vegetative and conjugative cells, with an increase at the poles during conjugation.

2. Fig 1C and H: It seems that these two panels were prepared with the same data set since the number of cells analyzed for each category are identical. It is however indicated that FigC corresponds to three biological replicates while Fig1H corresponds to one representative experiment. Please, clarify.

3. Figure 2 focuses on the ss-to-dsDNA conversion and the plasmid duplication. This figure is very informative and the data is convincing. It is unclear however which dataset was used to prepare the different panels. For example, the number of Ssb-Ypet conjugative foci analyzed before the appearance of the mCh-ParB focus is 342 (Fig 2B). Yet, the proportion of successful ss-to-DNA conversion in Figure 2C was calculated from 311 individual transfer events. Considering that the proportion of successful conversion is about 80%, I would expect the number of events analyzed in Fig2C to be above 400. On the contrary, the number of transconjugant analyzed in Fig2B (175) is lower than in Figure 2D (286). Finally, the number of Ssb-Ypet foci and the number of mCh-ParB foci before duplication analyzed in Figure 2B, 2E and 2F are different. The authors should maybe indicate in the figure legends or the Materials and Methods section why they did not use the same data sets to perform all the analyses (as they did with Figure 4 D, E, F).

4. Lines 221-239: The authors show that the number of F plasmids increases rapidly in transconjugant cells and reaches a final number of 4 copies per cell before division. Is there a correlation between the rate of plasmid duplication and the size (cell cycle stage) of the recipient cell?

Reviewer #2 (Remarks to the Author):

The manuscript reports on the use of state-of-the art fluorescence microscopy approaches to define the temporal sequence of events surrounding DNA transfer by the F plasmid conjugation system. The F system is an important model for understanding mechanistic and structural details of conjugation, which itself is enormously important for human medicine as a principle means of antibiotic resistance dissemination among bacteria. The authors have elegantly visualized the kinetics of accumulation of the single-stranded transfer intermediate (T-strand) in recipients, and

shown for the first time that ssDNA accumulates in donors as well during mating. They also define the kinetics of the ssDNA to dsDNA conversion, and kinetics of accumulation of proteins encoded by distinct regions of the F plasmid after transfer. The work is exceptionally well-developed and rigorous, and the findings both agree with and extend the previous literature on the F transfer system. Some assumptions are made that may or may not be correct. For example, the authors state that the entire F plasmid is transferred and the T-strand is circularized prior to lagging (second-strand) synthesis – this has never been shown to my knowledge. A second is that findings relating to the F system, or findings for other systems, are extrapolatable to all conjugation systems, which clearly is not the case. Specifically, here, the kinetics of tra gene expression is valid for this derepressed F system, but likely not for other repressed F plasmids or other conjugation systems relying on other regulatory cues for tra gene expression. A third is that the observed accumulation of ssDNA in the donor is a natural feature of conjugation – first, this might be observed in the rich media conditions used in laboratory matings and not in natural settings, second, the findings may not be extrapolatable to other conjugation systems, especially those relying on relaxases without associated helicases. These concerns and other comments listed below are readily addressed by inclusion of a few statements acknowledging the limitations of the findings and possible alternative explanations. Overall, the results are exciting and collectively represent an important advance for the field of conjugation/T4SS.

Comments:

1. Pg. 3. L. 49. Please explain the references to the Vir protein homologs – a sentence to the effect that the VirB/D4 proteins are considered archetypes of the core set of subunits required to build T4SSs, allowing for reference to a common nomenclature.
2. Pg. 4. L. 67. This has been termed the T-complex.
3. Pg. 4. Bottom. It remains unknown (to my knowledge) whether T-strand circularization by TraI occurs before or after lagging strand synthesis. Indeed, the findings presented here favor the notion that the incoming ssDNA is converted into dsDNA as it enters the recipient cell, since ssb-Ypet foci are seen at the mating junction. If circularization occurs prior to lagging strand synthesis, there's no reason for Ssb-Ypet foci to remain at the mating junction.
4. Pg. 5. L. 97. So far (including this study), F plasmids have been shown to carry only 1 or 2 ssDNA promoter sequences, one upstream of ssb (this study) and one downstream of the ssb/psiB/psiA gene cluster (previous work). This raises a question of whether leading region genes located 5' of these ssDNA promoters are actually expressed from ssDNA promoters. As a number of these genes (mostly hypothetical) are also highly conserved among F plasmids, the kinetics of their expression during mating should be investigated (in a future study).
5. P. 6. L. 120. This is the only copy of Ssb in these cells, correct? This should be stated.
6. Pg. 6. L. 132. Fig. S2A (third panel). These data in fact suggest that the Ssb-Ypet foci are enriched at the donor cell poles, not the quarter positions, which isn't consistent with the heat map presented in Fig. 1C (third panel) or what is stated in the text. Need to reconcile the different datasets.
7. Pg. 6. L. 130. Fig. S2B. These data support the idea that the incoming ssDNA remains at the mating junction rather than floating into the interior of the cell as would occur if the ssDNA were circularized prior to second strand synthesis. This implies that the replicative machinery might be fixed at the site of the incoming ssDNA for lagging strand synthesis.
8. Pg. 6. L. 136. The data don't really support such a conclusion. It'd be better to study gene transfer as a function of cell cycle using synchronized cell populations.
9. Pg. 7. L. 161. Fig. 1. This is the first reference to mCherry expression. Is this in the donor or recipient? Although some information is in the figure legend, it should be clarified here in the text.
10. Pg. 8. L. 178. Again, this assumes that the entire T-strand enters the recipient cell and

circularizes prior to lagging strand synthesis. The clustering of foci at the mating junction in recipient cells suggests that free circularized T-strands may not exist and that the replication machinery is recruited to the incoming ssDNA for lagging strand synthesis prior to recircularization. If so, the estimates for the amount of Ssb bound to incoming T-strand would be considerably lower than proposed.

11. Pg. L. 186. Would labeling of components of the replication machinery show redistribution to mating junctions as the T-strand enters the recipient cell?

12. Pg. 9. L. 211. The findings are highly interesting from the perspective of what is happening in the donor – it seems likely that multiple rounds of T-strand processing and transfer occur with a single F plasmid docked at TraD. The idea of successive RCR scrolling of T-strands into recipients is novel, I think.

13. Pg. 9. L. 213. Again, the alternative is that Okazaki fragments from lagging strand synthesis are being generated as the T-strand enters the cell. This would explain why everything – Ssb recruitment and ssDNA to dsDNA conversion (as marked by mCh-ParB recruitment) occur at the mating junction. According to such a model, the entire time-frame required for T-strand entry, dsDNA conversion and recircularization may be longer than the estimated 4 min.

14. Pg. 11. L. 256. Minor point – the gene fusions are to the 3' end of the leading region genes, not the C-terminal regions of these genes.

15. Pg. 12. L. 273. "raised" isn't the best word here, maybe "yielded"?

16. Pg. 12. L. 289. Fig. 3. S5. These data are hard to interpret – it seems from S5 that there isn't much (any?) difference between when the leading region protein fusions are first detected compared with the SopB or Tra protein fusions – a small number of cells show early production of all of these proteins prior to the 0 time (ParB focus detection). So, how was the diagram presented in Fig. 3B prepared?

17. Pg. 12. L. 285. Fig. 3B. All of the leading region genes analyzed are located between oriT and ssb. Given that these genes are all downstream of ssDNA promoters, the findings are reasonable (with the caveat of point 16 above). It would be interesting to repeat these analyses (in a future study) using fusions to leading region genes positioned between ssb and sopB. As no ssDNA promoters have yet been identified in this region, it seems likely that they would adopt the SopB profile, not those of the leading region genes analyzed here.

18. Pg. 13. L. 301. Fig. 3B. I was under the impression that new transconjugants can rapidly transfer F plasmids to new cells – within a time-frame that is much faster than the observed minimum of 20-30 min reportedly needed for synthesis of the Tra proteins. Do any data exist to this effect?

19. Pg. 14. L. 325. It would be relevant here to note that the F plasmid under study is derepressed for expression of the tra genes due to a mutation in FinO. Thus, the timing and production levels of the Tra proteins are relevant only for a derepressed F plasmid and not for other F plasmids with intact FinO/P repression systems, or indeed for other conjugation systems.

20. Pg. 14. L. 331. One caveat to this is that the leading region genes may be induced in donors upon establishment of the mating junction, which wasn't tested here.

21. Pg. 17. L. 420. TraN proteins encoded by other F plasmids bind other Omps. It would be interesting to determine if donors carrying those F plasmids show the same propensity to form mating junctions at the recipient cell poles. Alternatively, would a F plasmid deleted of traN show preferential mating junction formation at the recipient cell pole?

22. Pg. 17. L. 415. Neither of these references has anything to do with the F system, and extrapolation from the Agro or R388 systems to the F system may or may not be valid. Please cite another reference relating to F to make this point.

23. Pg. 18. L. 436. Could the accumulation of ssDNA in donors be an artefact of the experimental conditions employed, e.g., rich media mating conditions, and not really reflect what is happening in nature? If this scenario were applied to Hfr transfer, one would expect that within the time frame of an hour – which is required for transfer of the entire chromosome – that donors would accumulate an enormous amount of ssDNA. It'd be a simple test to see if Hfr donors accumulate an increasing amount of ssDNA over the 60 minute transfer period.

24. Pg. 18. L. 447. The underlying assumption here is that a copy of the relaxase remains in the donor to continue unwinding the T-strand from its template, while another copy of the relaxase pilots the T-strand into recipients. Since this has not been shown (to my knowledge), comparisons of different enzyme activity rates to account for accumulation of the ssDNA is highly speculative. Another point to consider is that many relaxases lack associated helicase domains. In such systems, would ssDNA also accumulate in donors?

25. Pg. 20. L. 485. The authors of that study proposed a two-stage model in which the early translocation of proteins might help to kickstart SOS suppression until gene expression from the ssDNA promoters yielded sufficient amounts of proteins to continue SOS suppression. In that context, it is noteworthy that this group also recently provided evidence for the translocation of other leading region proteins residing more distal to *ssb* – a region that appears to lack ssDNA promoters. The early translocation of those leading region proteins might supply functions for the window of time prior to ssDNA to dsDNA conversion that are important for biological roles yet to be defined (and perhaps detectable only under certain growth conditions and not in rich laboratory media). Anyway, the bottom line is that the results of Al Mamun and the present findings can be accommodated in a cohesive model. A sentence or two to this effect is warranted here.

26. Pg. 21. L. 511. It's relevant here to refer to the findings that translocation of *PsiB* and *Ssb* – and likely also the production of these proteins in new transconjugants – suppresses the mating-induced SOS response. This is an important example of a biological function for leading region proteins that is independent of any involvement in the DNA transfer process itself.

Reviewer #3 (Remarks to the Author):

The work by Couturier et al extensively uses live microscopy to monitor bacterial conjugation in order to define the spatiotemporal parameters of the process in vivo. The finely tuned microscopic experiments allow to visualize the conjugative process and to determine interesting parameters such as the entry of ssDNA in the recipient and concomitant generation of ssDNA in the donor, or the ss-to-dsDNA conversion in the recipient. This refined technique has been used by the authors in the past, and recently described in detail in a methodological journal.

The results obtained are consistent with current conjugation models, and observations are overwhelmingly logical, since:

- We would not expect SSB recruitment by the entering ssDNA to provoke a collapse of the DNA replication fork; if this were to happen, the in vivo consequences would have been detected long ago.
- The detection of multiple transfer events from the same donor is expected, considering that entry exclusion requires the expression of proteins encoded by the incoming DNA.
- The faster replication of the incoming plasmid until it reaches its average copy number is probably due to the nature of the copy number control systems.
- The timing of expression: genes after the ss-promoter are expressed first and transiently, while the others are expressed after ss-to dsDNA conversion. It is difficult to imagine it would have been the other way round.
- The expression of the different genes is controlled by the nature of their respective promoters, rather than by simply acquiring the dsDNA state (which would imply constitutive expression thereof).

The only results that had not been described/predicted by previous works, i.e. the presence of a second ss-promoter and the involvement of plasmid SSB in early plasmid duplication, do not seem to be physiologically relevant.

Thus, my major concern with this work is that I see no novelty in the results presented. It has been known for decades that the leading region carries ss-promoters for early expression in the recipient; what other purpose would they serve? They were shown to work as ssDNA promoters in vitro some 25 years ago.

So, while the technique is very interesting and may be useful to analyse and time the conjugation process in vivo, the results in this particular work do not provide any significant knowledge to the field. The technique itself has already been published on its own, and the results themselves are better suited for a more specialised journal.

Minor comments

- Sentence in ll 65-68 makes no sense... maybe change "If" by "While"?
- If donor ssDNA foci are all around and recipient foci are polar, shouldn't it be that most mating pairs would look perpendicular to one another, or correlative? Instead, it looks like most are aligned side by side, which contradicts the results of ssDNA polar foci in the transconjugants.
- L 356: less than one log drop in conjugation frequency is far from being a dramatic reduction.
- L 404: microscopy has been used to visualize conjugation for more than 20 years now, so "only recently" is not very accurate.
- L 426: why would it be unexpected that ssDNA appears first in the recipient cell? The current model proposes that the displacement of the transferred DNA strand is coupled with its translocation, while the remaining DNA strand is already primed for RCR replication; while in the recipient, newly entered ssDNA (not engaged to the cellular machinery) must undergo a ss-to-dsDNA conversion process. What is surprising to me is the appearance of ssDNA later in the donor. Considering that the same authors did not detect this ssDNA in the donor in their previous works, maybe they can discuss methodological details which could account for this difference, rather than challenging the current model.
- L 435, 438, please close brackets
- L 453-54, ss-dsDNA conversion is not the pivotal event controlling the plasmid gene expression, but only the on-off switch for the expression from ss-promoters (which is not an essential process for plasmid establishment in the recipient cell). As the authors show, the expression from the different ds promoters is (logically) driven by their own regulatory patterns.
- L 477 ff: protein transfer during conjugation was already shown by Wilkins more than 30 years ago (Rees and Wilkins 1990, PMID: 2172695)
- L488 until the end of discussion: the fact that mobilizable plasmids carrying only the oriT can be conjugatively transferred with 100% efficiency argues against a critical role of the leading region in conjugation.

REVIEWER COMMENTS

Reviewer #1 (Remarks to the Author):

The study by Couturier et al. develops a single cell approach to examine the timing and the subcellular location of key steps of the conjugation process. Overall, the manuscript provides significant insights into the fundamental field of bacterial conjugation and open new perspectives to study this mechanism.

The authors use different fluorescent tools to monitor the transfer of the ssDNA plasmid from the donor to the recipient cells, and to detect its conversion into dsDNA. They first show that plasmid ssDNA accumulates both in the recipient and donor cells and propose that the helicase unwinding activity of the dsDNA plasmid is faster than its replication by rolling circle mechanism. They demonstrate that multiple transfers occur within an established mating pair and that the ss-to-dsDNA conversion is spatially separated from plasmid duplication. Finally, they measured the production time window and the level of proteins synthesized by the plasmid in transconjugant cells. Their analysis provides evidence that the differences of production timing rely on the specificities of the promoters controlling the genes encoding these proteins.

The paper is well written, the data is assembled in a clear and concise way and conclusions in general are elegantly supported by experimental results and statistical analyses.

> We thank this reviewer for the constructive and positive evaluation of our work. We provide a point-by-point answer to his/her concerns below.

Minor comments:

1. Line 133, Fig S2A: It is written that there is a noticeable enrichment of the Ssb-Ypet foci at the cell quarter positions in the donor cells. The histograms shown in Fig S2A indicate however that the proportion of foci located at the quarter positions is similar in vegetative and conjugative cells, with an increase at the poles during conjugation.

> This reviewer is right. The data (Figure 1C, Figure S2A-B) show that Ssb-Ypet foci are peripheral and mostly quarter in conjugative donors, while they are inner yet equally quarter in vegetative donors. Our statement was ambiguous and misleading due to the inappropriate usage of the word “enrichment”. We simply meant to describe Ssb-Ypet localisation in conjugative cells rather than comparing to vegetative cells. Therefore, we modified the text to the simply descriptive statement: “Ssb conjugative foci are distributed at the periphery of the donor cell and preferentially at the quarter positions”.

2. Fig 1C and H: It seems that these two panels were prepared with the same data set since the number of cells analyzed for each category are identical. It is however indicated that FigC corresponds to three biological replicates while Fig1H corresponds to one representative experiment. Please, clarify.

> Thanks for noticing this error. In Fig. 1H, only the free mCherry data comes from one representative experiment, while indeed, other violin plots were prepared with the same data set than Fig 1C, i.e., from at least three biological replicates. This is now clearly stated in the legend: “Free mCherry data correspond to one representative experiment, while other violin plots correspond to the same data set than in panel (C) from at least three biological replicates. The number of cells analysed (n) is indicated”. We have also corrected the description of the strain at the end of the figure legend: “the free mCherry is produced from the chromosome in MS388 *wt* background (LY1737).”

3. Figure 2 focuses on the ss-to-dsDNA conversion and the plasmid duplication. This figure is very informative and the data is convincing. It is unclear however which dataset was used to prepare the different panels. For example, the number of Ssb-Ypet conjugative foci analyzed before the appearance of the mCh-ParB focus is 342 (Fig 2B). Yet, the proportion of successful ss-to-dsDNA conversion in Figure 2C was calculated from 311 individual transfer events. Considering that the proportion of successful conversion is about 80%, I would expect the

number of events analyzed in Fig2C to be above 400. On the contrary, the number of transconjugant analyzed in Fig2B (175) is lower than in Figure 2D (286). Finally, the number of Ssb-Ypet foci and the number of mCh-ParB foci before duplication analyzed in Figure 2B, 2E and 2F are different. The authors should maybe indicate in the figure legends or the Materials and Methods section why they did not use the same data sets to perform all the analyses (as they did with Figure 4 D, E, F).

> We understand that this brings some confusion (this is actually something we have discussed among the authors before submitting the manuscript), yet we confirm that all indicated sample size (n =) are accurate. So why are the sample size different between panels?

The first reason is that, because conjugation is asynchronous in the population, time lapse movies do not always capture the entire sequence of DNA transfer, i.e., Ssb-Ypet focus appearance/disappearance, mCh-ParB focus formation, 1st and 2nd mCh-ParB duplication events. In addition, using 1min/frame time lapses was suitable to analyse Ssb-Ypet appearance/disappearance relative to mCh-ParB formation (Fig. 2B-2E), but provoked mCh-ParB bleaching thus hindering the analysis of mCh-ParB duplication events. This is why we have used 5min/frame to analyse mCh-ParB 1st and 2nd duplication events relative to mCh-ParB focus formation (Fig. 2B and 2F). This is now clearly stated in panel 1B legend: “Ssb-Ypet focus appearance was analysed using 1min/frame time lapses, while mCh-ParB focus 1st and 2nd duplication events were analysed using 5min/frame time lapses.”

The second reason is that for the different transfer parameters we wanted to describe, we have gathered and analyse different sets of transfer events:

- Fig. 2B and Fig. 2C: To generate Fig. 2B, we have selected events showing mCh-ParB appearance (n = 342), and we have quantified the timing of Ssb-Ypet appearance in each frame. By contrast, to generate Fig. 2C, we have selected events showing Ssb-Ypet appearance/disappearance (n = 311), and we have quantified the proportion that shows the subsequent appearance of mCh-ParB (i.e., successful transfers).

- Fig. 2B and Fig. 2D: mCh-ParB duplication data in Fig. 1B (n= 175) come from 5min/frame time-lapses, while data presented in Fig. 2D come from 1min/frame time-lapses.

- Fig. 2E, 2F and Fig. 2B: Estimate of time lags only requires to count the number of frames between the two considered events (Ssb-Ypet appearance and disappearance in 2E; mCh-ParB focus formation and 1st or 2nd duplications in 2F). This analysis is a less demanding than for Fig. 2B, for which we had to annotate the presence/absence of Ssb-Ypet or mCh-ParB foci in each frame of the time lapses.

In summary, we realize that the indicated sample size needs to be explained better. We propose to leave these numbers as they are in the figure panels, while adding the above-mentioned statement in the figure legend, and the following text in the material and methods (L. 584-596): “Of note, using 1min/frame time lapses was suitable to analyse Ssb-Ypet appearance/disappearance relative to mCh-ParB formation (Fig. 2B-2E), but provoked mCh-ParB bleaching thus hindering the analysis of mCh-ParB duplication events on the long term. We have then used 5min/frame time lapses to analyse mCh-ParB 1st and 2nd duplication events relative to mCh-ParB focus formation (Fig. 2B and 2F). Also, the characterisation of the different transfer parameters was performed using specific analysis. For instance, time lags were calculated by counting the number of frames between the two considered events (Ssb-Ypet appearance and disappearance in Fig. 2E and 4E; mCh-ParB focus formation and 1st or 2nd duplications in Fig. 2F and 4F). By contrast, Fig. 2B and 4B were generated by annotating the presence/absence of Ssb-Ypet or mCh-ParB foci in each time lapse frame.”

Alternatively, and if requested by this reviewer or the editor, we can homogenise the indicated (n =) in the figure panel, which would require removing some data points.

4. Lines 221-239: The authors show that the number of F plasmids increases rapidly in transconjugant cells and reaches a final number of 4 copies per cell before division. Is there a correlation between the rate of plasmid duplication and the size (cell cycle stage) of the recipient cell?

> We also expected a higher plasmid replication rate in older (larger) transconjugants, to allow reaching ~4 plasmid copies before division. However, we do not observe any significant correlation between the cell length (age) of the transconjugant cell (at the time of ParB focus appearance) and time required for mCh-ParB duplication (Figure A1). To us, this points toward a second possibility, that cell division could be delayed (rather than plasmid replication accelerated) in older transconjugant cells compared to younger transconjugant cells. Supporting this view, we have quantified a significant increase in

generation time of transconjugant cells (not sorted by cell size) compared to recipients (and donors) (Figure A2). For this reviewer's information, we are currently developing a specific single-cell analysis method to investigating the interplay between the transconjugants' cell cycle and plasmid acquisition and dynamics (*and vice versa*).

We thank this reviewer for his/her interest in the work. We hope we provided satisfactory answers to his concerns.

Reviewer #2 (Remarks to the Author):

The manuscript reports on the use of state-of-the-art fluorescence microscopy approaches to define the temporal sequence of events surrounding DNA transfer by the F plasmid conjugation system. The F system is an important model for understanding mechanistic and structural details of conjugation, which itself is enormously important for human medicine as a principle means of antibiotic resistance dissemination among bacteria. The authors have elegantly visualized the kinetics of accumulation of the single-stranded transfer intermediate (T-strand) in recipients, and shown for the first time that ssDNA accumulates in donors as well during mating. They also define the kinetics of the ssDNA to dsDNA conversion, and kinetics of accumulation of proteins encoded by distinct regions of the F plasmid after transfer. The work is exceptionally well-developed and rigorous, and the findings both agree with and extend the previous literature on the F transfer system. Some assumptions are made that may or may not be correct. For example, the authors state that the entire F plasmid is transferred and the T-strand is circularized prior to lagging (second-strand) synthesis – this has never been shown to my knowledge. A second is that findings relating to the F system, or findings for other systems, are extrapolatable to all conjugation systems, which clearly is not the case. Specifically, here, the kinetics of *tra* gene expression is valid for this derepressed F system, but likely not for other repressed F plasmids or other conjugation systems relying on other regulatory cues for *tra* gene expression. A third is that the observed accumulation of ssDNA in the donor is a natural feature of conjugation – first, this might be observed in the rich media conditions used in laboratory matings and not in natural settings, second, the findings may not be extrapolatable to other conjugation systems, especially those relying on relaxases without associated helicases.

These concerns and other comments listed below are readily addressed by inclusion of a few statements acknowledging the limitations of the findings and possible alternative explanations. Overall, the results are exciting and collectively represent an important advance for the field of conjugation/T4SS.

> We thank this reviewer for his/her in depth and constructive assessment of our work. We have carefully considered all his/her comments, in particular regarding the assumption we (wrongly) made regarding the chronology between plasmid entry, circularisation and complementary strand synthesis. This subject is central to this reviewer's Points 3, 5, 7, 10, 13 and is key to the interpretation of our results. As this reviewer will appreciate in our response and in the revised manuscript, these comments led us to make several modifications in the text. The bottom line is that we totally agree that the timing of plasmid recircularization after entry is not known. We then modified the introduction accordingly. We also moderated some of our interpretations/statements accordingly. The second important point concerns the timing of complementary strand synthesis relative to plasmid entry. Does the ss-to-dsDNA conversion occur as the T-strand enters the cell or is it initiated after completion of plasmid entry? If our data do not provide any definitive answer, several results presented allow us to discuss this question in the point-by-point answer below and in our revised manuscript.

Comments:

1. Pg. 3. L. 49. Please explain the references to the Vir protein homologs – a sentence to the effect that the VirB/D4 proteins are considered archetypes of the core set of subunits required to build T4SSs, allowing for reference to a common nomenclature.

> We have added (L. 49-51): “TraI and TraD proteins are archetype components of the core set of subunits required for the establishment of an active conjugation machinery and are respectively referred to as VirD2 and VirD4 in the common nomenclature”.

2. Pg. 4. L. 67. This has been termed the T-complex.

> Indeed, thanks. The term “T-complex” was added as suggested.

3. Pg. 4. Bottom. It remains unknown (to my knowledge) whether T-strand circularization by TraI occurs before or after lagging strand synthesis. Indeed, the findings presented here favor the notion that the incoming ssDNA is converted into dsDNA as it enters the recipient cell, since *ssb*-Ypet foci are seen at the mating junction. If circularization occurs prior to lagging strand synthesis, there's no reason for *Ssb*-Ypet foci to remain at the mating junction.

> This comment invited us to “dig” into the literature for evidence that circularization occurs before (or after) complementary strand synthesis, and indeed, we find none. Several articles and reviews (PMID: ...) state that circularization occurs before complementary strand synthesis, but this was not supported

by reference of previous works clearly demonstrating that it is the case. So, we sincerely thank this reviewer for correcting this misconception. The introduction has been modified accordingly: (L. 72-78) “Once both 5’ and the 3’ ends of the T-strand have been internalised into the recipient cell, now called a transconjugant, the ssDNA plasmid is circularised by TraI. The ssDNA plasmid will also be converted into dsDNA by the complementary strand synthesis reaction^{26,27,51,52}. Whether this DNA synthesis reaction occurs as the plasmid enter the recipient cell or is initiated after plasmid recircularization remains unclear.”.

We also think it likely that after recircularization the plasmid would be free to move into the cytoplasm. However, none of our data (or published works) allow us exclude the possibility that the circularized plasmid would remain anchored to the mating junction (by some yet unknown mechanism). As a consequence, we are reluctant to interpret that the dwelling of Ssb-Ypet foci at the mating junction is a proof that the plasmid recircularization has not occurred yet (though we also favour this notion). Actually, it is fair to say that our data do not provide much information regarding the recircularization timing. The only thing we can say for sure is that recircularization has necessarily occurred when the *ygjA* gene is expressed (~15 min after mCh-ParB focus appearance), since *ygjA* promoter is on the other side of the *nic* site.

The second important aspect raised by this reviewer’s point concerns the timing of ss-to-dsDNA conversion relative to plasmid entry. This is further discussed in our answer to this reviewer’s Point 7, 10 and 13.

4. Pg. 5. L. 97. So far (including this study), F plasmids have been shown to carry only 1 or 2 ssDNA promoter sequences, one upstream of *ssb* (this study) and one downstream of the *ssb/psiB/psiA* gene cluster (previous work). This raises a question of whether leading region genes located 5’ of these ssDNA promoters are actually expressed from ssDNA promoters. As a number of these genes (mostly hypothetical) are also highly conserved among F plasmids, the kinetics of their expression during mating should be investigated (in a future study).

> We share this reviewer’s interest in the question “where does the leading region (in terms of early expressed genes) really start/stops”. This is a question we intend to address with the co-author Yoshiharu Yamaichi, whose work showed that the methylase MEcoGIX encoded by the pESBL plasmid is involved in the early steps of pESBL plasmid establishment in the recipient cell (PMID: 33270887). Interestingly, the F plasmid encodes the *yfeA* homologue gene, which is located precisely between the *SopABC* and the *Frpo2* loci, just upstream what is currently considered as the leading region. Analysis of *yfeA*-sfGFP production timing (together with other genes of the region) should provide answers regarding the boundaries of the leading region, and more generally the functions of leading factors in conjugation.

5. P. 6. L. 120. This is the only copy of Ssb in these cells, correct? This should be stated.

> The Ssb-Ypet fusion used throughout our study results from the insertion of the *ypet* gene in C-terminal of the *ssb* gene at the endogenous chromosome locus. This fusion was originally constructed by lambda red recombination and reported in Reyes-Lamothe et al., 2008 (PMID: 18394992). The word “endogenous” has been added in the text. (L. 105)

6. Pg. 6. L. 132. Fig. S2A (third panel). These data in fact suggest that the Ssb-Ypet foci are enriched at the donor cell poles, not the quarter positions, which isn’t consistent with the heat map presented in Fig. 1C (third panel) or what is stated in the text. Need to reconcile the different datasets.

> This reviewer is right and this ambiguity was also raised by reviewer1 (Please also read our answer to reviewer1’s Point1 above). The word “enriched” was inappropriate and has been removed as our intention was to describe Ssb-Ypet localisation in transconjugant cells rather than compare it to donor cells. The text now reads (L. 117-118): “Ssb conjugative foci are distributed along the donor cells’ side and preferentially at the cell quarter positions”.

7. Pg. 6. L. 130. Fig. S2B. These data support the idea that the incoming ssDNA remains at the mating junction rather than floating into the interior of the cell as would occur if the ssDNA were circularized prior to second strand synthesis. This implies that the replicative machinery might be fixed at the site of the incoming ssDNA for lagging strand synthesis.

> We agree that the close positions of appearance of Ssb-Ypet and mCh-ParB foci indicates that the ss-to-dsDNA replication machinery is recruited onto the plasmid when it is still associated with the mating junction. This is stated (L. 235-238): “The recruitment of the complementary strand synthesis machinery and the ss-to-dsDNA replication reaction occur in the vicinity of the polar position of entry of the ssDNA plasmid”. However, as indicated in our answer to this reviewer’s point3 and consistent with Fig. S2B, we do not think our data allow to validate the (likely) assumption that the plasmid DNA would be free to float in the cell’s cytoplasm after circularisation (rather than staying attached by some undescribed interaction).

8. Pg. 6. L. 136. The data don’t really support such a conclusion. It’d be better to study gene transfer as a function of cell cycle using synchronized cell populations.

> Indeed, using synchronized cell populations would be best experiment to address whether conjugation is strictly independent of the cell cycle stage. Still, we think that the broad cell size distribution of donor and recipient engaged in conjugation is a very strong clue that it is the case. We have nonetheless moderated our conclusion in the revised text by removing “conjugation is cell-cycle independent”. The text now reads (L. 124-125): “This shows that the donors can give, and recipients can acquire the plasmid at any stage of their cell cycle, from birth to cell division.”, which is an accurate description of our results.

9 Pg. 7. L. 161. Fig. 1. This is the first reference to mCherry expression. Is this in the donor or recipient? Although some information is in the figure legend, it should be clarified here in the text.

> Clarification about this strain was also requested by reviewer1’s (please see our response to his/her point2). The mCherry skewness was analysed in the wild type MS388 F- strain producing the mCherry from a constitutive promoter on the chromosome (strain LY1737). This is our reference strain (MG1655 StR) that was used both as recipient (without or without fluorescent reporters), or as donor (when carrying the F plasmid, without or without fluorescent reporters) in this study. We have clarified this in Figure 1 legend: “the free mCherry is produced from the chromosome in MS388 *wt* background (LY1737).”

10. Pg. 8. L. 178. Again, this assumes that the entire T-strand enters the recipient cell and circularizes prior to lagging strand synthesis. The clustering of foci at the mating junction in recipient cells suggests that free circularized T-strands may not exist and that the replication machinery is recruited to the incoming ssDNA for lagging strand synthesis prior to recircularization. If so, the estimates for the amount of Ssb bound to incoming T-strand would be considerably lower than proposed.

> We agree with this reviewer that “that free circularized T-strands may not exist” and by extension, that coating of the 108 000 nucleotides ssDNA F plasmid by Ssb might never be required. To account for this gap in knowledge and moderate our interpretation, we have added a statement in this section (L.165-169): “In fact, it is not known whether the F plasmid is ever fully present in ssDNA form in the recipient, as it is not known if the complementary strand synthesis reaction occurs concomitantly with or after completion of the T-strand internalisation. Still, the observe massive recruitment of Ssb molecules onto the incoming ssDNA could reduce Ssb availability and provoke a transitory disturbance of the host chromosome DNA replication.”

11. Pg. L. 186. Would labeling of components of the replication machinery show redistribution to mating junctions as the T-strand enters the recipient cell?

> Thanks for this suggestion. Indeed, our data lead us to propose that the replication machinery required for ss-to-dsDNA transfer is recruited onto the incoming ssDNA. This recruitment could be revealed using functional fluorescent fusions of replisome components, which are available thanks to the work of the Reyes LAB and others. One limitation might be the (too small) number of molecules recruited to the plasmid that might challenge their localisation, in which case single-particle tracking might be strategic.

12. Pg. 9. L. 211. The findings are highly interesting from the perspective of what is happening in the donor – it seems likely that multiple rounds of T-strand processing and transfer occur with a single F plasmid docked at TraD. The idea of successive RCR scrolling of T-strands into recipients is novel, I think.

> Thanks for emphasizing the novelty of this finding.

13. Pg. 9. L. 213. Again, the alternative is that Okazaki fragments from lagging strand synthesis are being generated as the T-strand enters the cell. This would explain why everything – Ssb recruitment and ssDNA to dsDNA conversion (as marked by mCh-ParB recruitment) occur at the mating junction. According to such a model, the entire time-frame required for T-strand entry, dsDNA conversion and recircularization may be longer than the estimated 4 min.

> As indicated in our answers to this reviewer's point 3, 7 and 10, we agree our data do not to conclude regarding the timing of recircularization relative to ss-to-dsDNA conversion. As a consequence, we modified our statement by removing the part "the circularisation of the ssDNA plasmid by TraI".

Interpretation of our data regarding the timing of complementary strand synthesis relative to plasmid entry is more arduous. Indeed, the *parS* site is inserted only 647 pb upstream the *nic* site of the *oriT* and is, consequently, among the first region to enter the recipient. If the ssDNA was converted into dsDNA as the T-strand enters the cell, the *parS* site would be among the first sequence to be converted into dsDNA, and the appearance of the ParB focus would reflect the primary step of ss-to-dsDNA conversion. In this view, the mCh-ParB focus would colocalize with the Ssb-Ypet until completion of ssDNA transfer and ss-to-dsDNA conversion (basically, as long as the plasmid is a mix of both ssDNA and dsDNA form). Yet, our 1 min/frame time-lapses show that the Ssb-Ypet focus generally disappears before mCh-ParB focus formation. One first possibility is that the ss-to-dsDNA conversion is completed in a minute-scale period of time, and is therefore probably patchy from multiple initiation sites rather than continuous from a single initiation origin (continuous replication of the 108 kb F plasmid would take ~155 second assuming a 700 nuc.s⁻¹ replication rate). Another possibility is that our *parS* site is not among the first region to be converted into ss-to-dsDNA. Looking at different *parS* site insertions and increase the temporal resolution of our experimental approach (below 1 min/frame) would allow addressing this question.

As a consequence, we realize we cannot say that the ~4 min period of time encompasses completion of ss-to-dsDNA conversion. We have then moderated our conclusion of this paragraph (L202-205): "This period reflects the time required for the completion of a reaction cascade that comprises the internalisation of the ssDNA plasmid, the initiation of the complementary strand synthesis replication, and the recruitment of ParB molecules on the *parS* site in dsDNA form."

14. Pg. 11. L. 256. Minor point – the gene fusions are to the 3' end of the leading region genes, not the C-terminal regions of these genes.

> This has been modified. (L. 243)

15. Pg. 12. L. 273. "raised" isn't the best word here, maybe "yielded"?

> This has been modified. (L.260)

16. Pg. 12. L. 289. Fig. 3. S5. These data are hard to interpret – it seems from S5 that there isn't much (any?) difference between when the leading region protein fusions are first detected compared with the SopB or Tra protein fusions – a small number of cells show early production of all of these proteins prior to the 0 time (ParB focus detection). So, how was the diagram presented in Fig. 3B prepared?

> Fig. S5 top panels present the raw data of intracellular sfGFP fluorescence signal (automatically extracted by MicrobeJ), which reflect an equilibrium between production and loss (degradation probably) of the fusions. These data sets have been translated into fold-increase histograms (below each graph), which present the ratio (sfGFP signal at $T_{(n+10\text{min})}$ /sfGFP signal at T_n) where n is the time point. Fold-increase quantifies the variation in intracellular sfGFP fluorescence by 10 min sliding windows, where fold-increase > 1 indicates that the protein has been produced in the corresponding time interval. These graphs show that leading proteins (green graphs) start being produced before T_0 ; SopB (blue graph) between T_0 and T_{10} ; TraM between T_0 and T_{10} (similar to SopB indeed), TraC between T_{10} and T_{20} , and TraS and TraT between T_{20} and T_{30} . The polygonal shapes (fold-increase ≥ 1) on each of these graphs have been all reported on a single timeline to generate the Graph Fig. 3B (fusions production as a function of time). We realise these graphs maybe hard to interpret, but we think it is important to show the raw data. This is why we have tried to integrate/digest these data in one and only Figure 3B, as the coast of some information loss (mainly, Fig. 3B ignores the absolute protein level and do not show that some proteins are produced at higher levels than others).

17. Pg. 12. L. 285. Fig. 3B. All of the leading region genes analyzed are located between *oriT* and *ssb*. Given that these genes are all downstream of *ssDNA* promoters, the findings are reasonable (with the caveat of point 16 above). It would be interesting to repeat these analyses (in a future study) using fusions to leading region genes positioned between *ssb* and *sopB*. As no *ssDNA* promoters have yet been identified in this region, it seems likely that they would adopt the *SopB* profile, not those of the leading region genes analyzed here.

> We agree with this reviewer that our data raises the question, how far from the *oriT* are the early expressed genes? what is the gene expression regulation of genes upstream of *Frho2*? Also, see our answer to this reviewer's Point4.

18. Pg. 13. L. 301. Fig. 3B. I was under the impression that new transconjugants can rapidly transfer F plasmids to new cells – within a time-frame that is much faster the observed minimum of 20-30 min reportedly needed for synthesis of the *Tra* proteins. Do any data exist to this effect?

> This is another question we also would like to address in the close future: “how fast can a new transconjugant give the plasmid to another recipient”. In our microfluidics time-lapse movies, we do observe plasmid transfer from fresh transconjugants to neighbouring recipients. However, analysing this phenomenon will require accumulating more data by imaging the conjugation mix for a longer period of time, and potentially in a medium richer than M9CaaGluc (to accelerate protein production and T to R conjugation events). However, we have not performed this specific analysis yet so we cannot comment on what has been proposed in previous works. In the present work, we limited the analysis to transfer from donors to recipients only.

19. Pg. 14. L. 325. It would be relevant here to note that the F plasmid under study is derepressed for expression of the *tra* genes due to a mutation in *FinO*. Thus, the timing and production levels of the *Tra* proteins are relevant only for a derepressed F plasmid and not for other F plasmids with intact *FinO/P* repression systems, or indeed for other conjugation systems.

> That is right, we should have mentioned this important point. This is done in the revised version on the manuscript. (L. 310-314) :”It is important to stress that the F plasmid carries an naturally occurring insertion of the IS3 insertion sequence into the *finO* gene of the *FinOP* fertility inhibition system, which results in the upregulation and constitutive expression of *tra* genes⁷³. Therefore, other IncF plasmids in which the *FinOP* regulatory system is still active are expected to exhibit different timings and production levels of the *Tra* proteins than reported here.”

20. Pg. 14. L. 331. One caveat to this is that the leading region genes may be induced in donors upon establishment of the mating junction, which wasn't tested here.

> Actually, we have analysed sfGFP fusions levels in the donors during conjugation. The graph below show that the intracellular levels of leading proteins is not increased after 1 hour of conjugation (t1H) compared to donors before conjugation mix (t0). We can include this data in the supplementary if requested by this reviewer or the editor.

21. Pg. 17. L. 420. TraN proteins encoded by other F plasmids bind other Omps. It would be interesting to determine if donors carrying those F plasmids show the same propensity to form mating junctions at the recipient cell poles. Alternatively, would a F plasmid deleted of traN show preferential mating junction formation at the recipient cell pole?

> Thanks for these suggestions that would allow testing the hypothesis we put forward to explain F polar entry into the recipient cells. Indeed, the recent work by Wen Low et al., (PMID: 35697796) provide insights into specific interaction between TraN variants of different plasmids and outer membrane protein from different recipients. This could help designing experiments to address the influence of mating pair stabilisation systems on plasmid entry localisation.

22. Pg. 17. L. 415. Neither of these references has anything to do with the F system, and extrapolation from the Agro or R388 systems to the F system may or may not be valid. Please cite another reference relating to F to make this point.

> Unfortunately, we do not know of any published work reporting the localisation of T4SS components encoded by the F plasmid. This is why we cited these references to pTi and R388 plasmid systems, but we agree the parallel might not be valid. We then added a statement in the revised manuscript (L. 400-404): “This pattern could reflect the intracellular position of the T4SS machinery of the F plasmid, which to our knowledge, remains to be described. This possibility would be weakened if the F plasmid T4SS machinery are homogeneously located throughout the periphery of the cells as in the case of the pTi and R388 plasmids^{80, 82}. Alternatively, the lateral localisation of active conjugation pores may reflect the facilitated access to F plasmid molecules, which are also positioned at quarter positions and excluded from the cell poles^{84,85}.”

23. Pg. 18. L. 436. Could the accumulation of ssDNA in donors be an artefact of the experimental conditions employed, e.g., rich media mating conditions, and not really reflect what is happening in nature? If this scenario were applied to Hfr transfer, one would expect that within the time frame of an hour – which is required for transfer of the entire chromosome – that donors would accumulate an enormous amount of ssDNA. It'd be a simple test to see if Hfr donors accumulate an increasing amount of ssDNA over the 60 minute transfer period.

> For this reviewer's information, we also observe ssDNA accumulation in poorer medium (M9 Gluc), but this does not tell us much about what really happens during conjugation in natural environment, indeed. Besides, we totally agree that visualization of Hfr transfer would provide many answers and allow to challenge our current model. This is a follow-up project that we are only initiating in the lab and we wish we had answers already. Yet, we really appreciate this reviewer's interest and take it as an encouragement for our future works.

24. Pg. 18. L. 447. The underlying assumption here is that a copy of the relaxase remains in the donor to continue unwinding the T-strand from its template, while another copy of the relaxase pilots the T-strand into recipients. Since this has not been shown (to my knowledge), comparisons of different enzyme activity rates to account for accumulation of the ssDNA is highly speculative. Another point to consider is that many relaxases lack associated helicase domains. In such systems, would ssDNA also accumulate in donors?

> The transfer of the relaxase in the recipient and its activity in plasmid recircularization have been convincingly supported by the work Dóstal et al., 2011 (PMID: 21109533). The existence of the two relaxase conformers performing the helicase or the nicking activity in the donor are supported by Ilangovan et al., 2017 (PMID: 28457609). However, data are lacking regarding the coordination of these activities in the donor and recipient cells. We think this is one of the aspect that our live-cell approach allow to investigate. In this context, we agree that the comparison of enzymes rate estimated from different works is speculative and the discussion of our results implies some level of assumption. For this reason, we have tried to write the discussion avoiding overstatements by using conditional and words such as “support the view” “it is possible that” to make clear that we are proposing an interpretation that reconciles our results and the literature, rather than trying to push a model.

25. Pg. 20. L. 485. The authors of that study proposed a two-stage model in which the early translocation of proteins might help to kickstart SOS suppression until gene expression from the ssDNA promoters yielded sufficient amounts of proteins to continue SOS suppression. In that context, it is noteworthy that this group also recently provided evidence for the translocation of other leading region proteins residing more distal to ssb – a region that appears to lack ssDNA promoters. The early translocation of those leading region proteins might

supply functions for the window of time prior to ssDNA to dsDNA conversion that are important for biological roles yet to be defined (and perhaps detectable only under certain growth conditions and not in rich laboratory media). Anyway, the bottom line is that the results of Al Mamun and the present findings can be accommodated in a cohesive model. A sentence or two to this effect is warranted here.

> We agree that there is no intrinsic conflict between our data and the findings in the above-mentioned works. This is what we have tried to put forward in the manuscript. Nonetheless, following this reviewer's suggestion, we have written an additional statement to make it even clearer in the revised version (L475-478): "Therefore, it appears likely that direct protein translocation and *de novo* synthesis are concomitant mechanisms ensuring the presence of leading factors immediately upon entry of the ssDNA plasmid in the transconjugant cell."

26. Pg. 21. L. 511. It's relevant here to refer to the findings that translocation of PsiB and Ssb – and likely also the production of these proteins in new transconjugants – suppresses the mating-induced SOS response. This is an important example of a biological function for leading region proteins that is independent of any involvement in the DNA transfer process itself.

> Thanks for this suggestion, it is important indeed to mention that leading factors could play a role in the transconjugant physiology rather than in plasmid transfer *per se*. This would also be beneficial for the overall plasmid dissemination in bacterial population at longer time scales. We have added the statement (L. 496-499) :“Yet, Ssb and PsiB have been shown to suppress conjugation-induced SOS induction in the transconjugant cell (REF), which is likely important for the transconjugant's physiology and proliferation rather than for plasmid transfer *per se*.”

We thank this reviewer for his/her scientific interest and the question it raises for future studies. We also appreciate his/her constructive comments, which helped us improving the quality and accuracy of our manuscript.

Reviewer #3 (Remarks to the Author):

The work by Couturier et al extensively uses live microscopy to monitor bacterial conjugation in order to define the spatiotemporal parameters of the process *in vivo*. The finely tuned microscopic experiments allow to visualize the conjugative process and to determine interesting parameters such as the entry of ssDNA in the recipient and concomitant generation of ssDNA in the donor, or the ss-to-dsDNA conversion in the recipient. This refined technique has been used by the authors in the past, and recently described in detail in a methodological journal.

The results obtained are consistent with current conjugation models, and observations are overwhelmingly logical, since:

- We would not expect SSB recruitment by the entering ssDNA to provoke a collapse of the DNA replication fork; if this were to happen, the *in vivo* consequences would have been detected long ago.
- The detection of multiple transfer events from the same donor is expected, considering that entry exclusion requires the expression of proteins encoded by the incoming DNA.
- The faster replication of the incoming plasmid until it reaches its average copy number is probably due to the nature of the copy number control systems.
- The timing of expression: genes after the ss-promoter are expressed first and transiently, while the others are expressed after ss-to dsDNA conversion. It is difficult to imagine it would have been the other way round.
- The expression of the different genes is controlled by the nature of their respective promoters, rather than by simply acquiring the dsDNA state (which would imply constitutive expression thereof).

We thank this reviewer for evaluating our work and for appreciating the relevance of the above listed observations, and their consistency with current conjugation models.

The only results that had not been described/predicted by previous works, i.e. the presence of a second ss-promoter and the involvement of plasmid SSB in early plasmid duplication, do not seem to be physiologically relevant.

We guess that by “physiologically relevant”, this reviewer means “critical to conjugation efficiency.”. If so, this reviewer is right that *ssb^F* or *Frpo2* deletions do not impact conjugation efficiency in the condition tested. However, it cannot be excluded that the relevance of these genes might be better revealed during conjugation in less favourable conditions, or between more phylogenetically distant bacterial species (as we state at the end of the discussion section). Besides, genes like *psiB* and *ssb* suppress conjugation-induced SOS response in the transconjugants, which is more important for the physiology of the transconjugant cell than for plasmid transfer efficiency *per se*, yet still “physiologically relevant” (see our answer to reviewer2’s Point26).

Thus, my major concern with this work is that I see no novelty in the results presented.

It has been known for decades that the leading region carries ss-promoters for early expression in the recipient; what other purpose would they serve? They were shown to work as ssDNA promoters *in vitro* some 25 years ago.

We find questionable that this reviewer uses a rhetorical argument such as “what other purpose would they serve?” to undermined the novelty of the results presented. Using rational logic to propose that something should happen is usually considered as a hypothesis rather than a scientific proof. In this particular case, as indicated by this reviewer, ssDNA promoters have been (indeed) proposed from *in vitro* works (that we cite) but our present work demonstrate that they actually act as so during *in vivo* conjugation. In addition, we also report for the first time the quantitative description of their role on the expression of leading genes.

So, while the technique is very interesting and may be useful to analyse and time the conjugation process *in vivo*, the results in this particular work do not provide any significant knowledge to the field. The technique itself has already been published on its own, and the results themselves are better suited for a more specialised journal.

We are confused as “analyse and time the conjugation process *in vivo*” is exactly what we used our technique for in the present study. None of our data regarding the chronology and intracellular localisation of conjugation step have been reported before.

Minor comments

- Sentence in ll 65-68 makes no sense... maybe change “If” by “While”?

This has been modified (L. 59).

- If donor ssDNA foci are all around and recipient foci are polar, shouldn't it be that most mating pairs would look perpendicular to one another, or correlative? Instead, it looks like most are aligned side by side, which contradicts the results of ssDNA polar foci in the transconjugants.

After cell loading in the microfluidic chamber, we obtain a mono-layer of cells in which the position of the donors and recipients relative to one another is random. Our results reflect the analysis of hundreds of mating pairs within these microscopy fields where cells are at confluence. For the sake of clarity, the figures presenting plasmid transfer dynamics show isolated mating pairs where the relative position of the cells relative to one another might not be representative (we favoured the representativity of the transfer event but raw data are made available with the submission of our work).

- L 356: less than one log drop in conjugation frequency is far from being a dramatic reduction.

In the revised manuscript, we have written “significantly” instead of “dramatically” (L. 345).

- L 404: microscopy has been used to visualize conjugation for more than 20 years now, so “only recently” is not very accurate.

Recently might be relevant in this case, as conjugation is being investigated since the 1946.

- L 426: why would it be unexpected that ssDNA appears first in the recipient cell? The current model proposes that the displacement of the transferred DNA strand is coupled with its translocation, while the remaining DNA strand is already primed for RCR replication; while in the recipient, newly entered ssDNA (not engaged to the cellular machinery) must undergo a ss-to-dsDNA conversion process. What is surprising to me is the appearance of ssDNA later in the donor. Considering that the same authors did not detect this ssDNA in the donor in their previous works, maybe they can discuss methodological details which could account for this difference, rather than challenging the current model.

We are confused by this contradictory comment, as this reviewer says “why would it be unexpected that ssDNA appears first in the recipient cell” and he/she also says “What is surprising to me is the appearance of ssDNA later in the donor”. The detection of ssDNA in the donor is new indeed. This reviewer refers to our previous article Nolivos et al., (PMID: 31123134). Indeed, we did not report ssDNA in the donor in this previous work, for the simple reason that we did not visualise Ssb-Ypet reporter in the donor strain at all (only in the recipient).

- L 435, 438, please close brackets

This has been done.

- L 453-54, ss-dsDNA conversion is not the pivotal event controlling the plasmid gene expression, but only the on-off switch for the expression from ss-promoters (which is not an essential process for plasmid establishment in the recipient cell). As the authors show, the expression from the different ds promoters is (logically) driven by their own regulatory patterns.

As explained throughout our manuscript, the ss-dsDNA conversion is the event that switches from the expression of leading genes (before ss-to-dsDNA conversion) to the expression of other genes controlled by dsDNA promoter (after ss-to-dsDNA conversion). Therefore, we think the statement “ss-dsDNA conversion is not the pivotal event controlling the plasmid gene expression” is accurate.

- L 477 ff: protein transfer during conjugation was already shown by Wilkins more than 30 years ago (Rees and Wilkins 1990, PMID: 2172695)

This reviewer is right, but in this section, we only refer to a specific set of plasmid-encoded proteins, which justify the reference to Al Mamun et al., 2021.

- L488 until the end of discussion: the fact that mobilizable plasmids carrying only the oriT can be conjugatively transferred with 100% efficiency argues against a critical role of the leading region in conjugation. This reviewer is right that the leading region is not required for the transfer of mobilizable plasmids.

However, all the work we cite after the statement “suggest a critical role of the leading region in conjugation” refer to autonomous conjugative plasmids, which are very different from mobilizable plasmids (much larger in size and likely in DNA metabolism needs, and in cost for the recipient cell). In addition to this previous works, our data also report that some deletions in leading region critically reduce conjugation efficiency. We meant “critical” and not “essential”.

We thank this reviewer for evaluating our work.

Reviewer #1 (Remarks to the Author):

The authors have adequately addressed my questions. I am also satisfied with the modifications made to the text in general. I recommend this manuscript for publication in Nature Communications.

Typo errors:

- Ln 72: Once both the 5' and 3' ends
- Ln 168: the observed massive recruitment
- Ln 330: early and transient expression
- Ln 347: conjugation efficiency
- Ln 402: the F plasmid T4SS machinery is homogenously located
- Ln 495: reference is missing

Reviewer #2 (Remarks to the Author):

This manuscript remains an elegant study that has confirmed but also significantly extended previous findings, many of which were made many years using biochemical or genetic techniques. The importance of this work rests not only in the data presented and new mechanistic information gained, but also in the advancement of relatively high resolution state-of-the-art fluorescence microscopy for visualization and detailed kinetic studies of the mating process. Although fluorescence microscopy has been applied to study some aspects of conjugation in the past, with the exception of the elegant work by Silverman and Clarke on the dynamics of F pilus assembly/retraction, no other studies have carefully detailed dynamic processes associated with DNA transfer among bacteria using single-cell live imaging. The authors have satisfactorily addressed all of my previous concerns and, I believe, those of the other reviewers. The findings are exciting, and they offer a glimpse of future applications of this powerful approach for addressing many other outstanding questions in the field of conjugation.

REVIEWERS' COMMENTS

Reviewer #1 (Remarks to the Author):

The authors have adequately addressed my questions. I am also satisfied with the modifications made to the text in general. I recommend this manuscript for publication in Nature Communications.

Typo errors:

Ln 72: Once both the 5' and 3' ends

Ln 168: the observed massive recruitment

Ln 330: early and transient expression

Ln 347: conjugation efficiency

Ln 402: the F plasmid T4SS machinery is homogenously located

Ln 495: reference is missing

> We thank this reviewer for his comments.

We have made all the suggested changes in the revised version of our manuscript.

Reviewer #2 (Remarks to the Author):

This manuscript remains an elegant study that has confirmed but also significantly extended previous findings, many of which were made many years using biochemical or genetic techniques. The importance of this work rests not only in the data presented and new mechanistic information gained, but also in the advancement of relatively high resolution state-of-the-art fluorescence microscopy for visualization and detailed kinetic studies of the mating process. Although fluorescence microscopy has been applied to study some aspects of conjugation in the past, with the exception of the elegant work by Silverman and Clarke on the dynamics of F pilus assembly/retraction, no other studies have carefully detailed dynamic processes associated with DNA transfer among bacteria using single-cell live imaging. The authors have satisfactorily addressed all of my previous concerns and, I believe, those of the other reviewers. The findings are exciting, and they offer a glimpse of future applications of this powerful approach for addressing many other outstanding questions in the field of conjugation.

> We thank this reviewer for his comments.